# A Highly Effective African Swine Fever Virus Vaccine Elicits a Memory T Cell Response in Vaccinated Swine

**DOI:** 10.3390/pathogens11121438

**Published:** 2022-11-29

**Authors:** Sarah E. Attreed, Christina Silva, Sophia Abbott, Elizabeth Ramirez-Medina, Nallely Espinoza, Manuel V. Borca, Douglas P. Gladue, Fayna Diaz-San Segundo

**Affiliations:** 1U.S. Department of Agriculture, Agricultural Research Service, Plum Island Animal Disease Center, Greenport, New York, NY 11944, USA; 2PIADC Research Participation Program, Oak Ridge Institute for Science and Education (ORISE), Oak Ridge, TN 37830, USA

**Keywords:** African Swine Fever Virus, live attenuated virus, vaccine, memory T cell, cellular immune response, ASF, vaccine, swine, I177L

## Abstract

African Swine Fever Virus (ASFV) is the causative agent of a highly contagious and lethal vector-borne disease in suids. Recently, a live attenuated virus strain, developed using the currently circulating, virulent Georgia strain (ASFV-G) with a single gene deletion (ASFV-G-ΔI177L), resulted in an effective vaccine. Nevertheless, protective immune response mechanisms induced by this candidate are poorly understood. In this study, Yorkshire crossbred swine intramuscularly vaccinated with 10^6^ 50% hemadsorption dose (HAD_50_) of ASFV-G-ΔI177L or a vehicle control were challenged at 28 days post-inoculation (dpi) with 10^2^ HAD_50_ of ASFV-G. Analysis of purified peripheral blood mononuclear cells following inoculation and challenge revealed that CD4+, CD8+ and CD4+CD8+ central memory T cells (CD44+CD25−CD27−CD62L+CCR7+, T_cm_) decreased significantly by 28 dpi in ASFV-G-ΔI177L-vaccinated swine compared to baseline and time-matched controls. Conversely, CD4+, CD8+ and CD4+CD8+ effector memory T cells (CD44+CD25−CD27−CD62−CCR7−, T_em_) increased significantly among ASFV-G-ΔI177L-vaccined swine by 28 dpi compared to baseline and time-matched controls. Additionally, the percentage of natural killer (NK), CD4+ and CD4+CD8+ T_em_ and CD8+ T_cm_ and T_em_ positive for IFNγ increased significantly following inoculation, surpassing that of controls by 28 dpi or earlier. These results suggest that NK and memory T cells play a role in protective immunity and suggest that studying these cell populations may be a surrogate immunity marker in ASF vaccination.

## 1. Introduction

African Swine Fever (ASF) is a highly contagious, often lethal hemorrhagic disease of both domestic and wild suids. ASF virus (ASFV), a large double-stranded DNA arbovirus and the only member of the family Asfarviridae, is the causative agent of this disease, infecting mononuclear phagocytic cells. ASF was first characterized in Western scientific literature in Kenya in 1921 [1], it is currently endemic in almost two dozen African countries, and it has been reported in 32 countries on the African continent since 2005 [2]. ASFV infection follows a sylvatic cycle, transmitting between agricultural populations of hogs and wild reservoirs of warthogs and ticks of the genus *Ornithodoros*. Over the 20th Century, several outbreaks of ASF were reported in Europe, East Asia and the Dominican Republic with devastating economic impacts. In 2007, an accidental introduction of ASFV into the Republic of Georgia resulted in a wide-scale epidemic throughout eastern Europe and Russia of a highly virulent strain known as the Georgia strain (ASFV-G) [3]. This epidemic has since touched off a fast-moving outbreak in Southeast Asia in early 2019 [4,5] and an outbreak in the Dominican Republic in the summer of 2021, followed by Haiti reporting presence of the disease to the World Organization for Animal Health (OIE) a few months later [6]. This latest outbreak on the island of Hispaniola is the first time in 40 years that ASF has been documented in the Western hemisphere, elevating the disease to the level of a global pandemic.

Due to the complex series of reservoirs for ASFV, eradication is not currently a feasible means of disease control. Instead, vaccines would be the preferred method of control, though a successful commercial vaccine has been, until recently, elusive. Recently, a live attenuated vaccine developed by introducing a single gene deletion of I177L in the currently circulating ASFV-G strain was found to be highly effective at preventing clinical disease in animals later challenged with wild type orthologous virus [7,8] as well as a currently circulating field strain in Southeast Asia [9]. However, the immunological correlates of protection are not perfectly understood for this or other naturally attenuated ASFV strains. Protection induced by passive transfer of hyperimmune serum from convalescent to naive animals [10] as well as in vitro naïve PBMCs [11] suggests a role for neutralizing antibodies in protection. Recently, studies of the immune response induced by naturally attenuated ASFV strains correlated protection with increased numbers of interferon (IFN)γ producing cells [12], although this correlation was not always observed [13,14]. Furthermore, depletion of CD8+ cells using specific antibodies diminished protection, demonstrating the importance of the cellular response in protection against ASF [15]. Besides, the innate immune response has also been described to take a part in protection against ASF. Leitão et al. showed how protection induced by attenuated NH/P68 isolate correlated with an early increase in Natural Killer (NK) cells activation [16]. On the other hand, although preliminary data indicated that ASFV developed mechanisms to escape dendritic cells (DC) defenses [17] little has been described about correlation between DCs and protection against ASF.

In the present study, ASFV-G-ΔI177L-vaccinated swine were challenged with the orthologous wild type virus at 28 dpi and samples were collected over the course of the experiment to assess humoral and cellular immune responses. Parameters were compared with baseline data and also with time-matched placebo control animals. Results indicate that not only does the ASFV-G-ΔI177L vaccine elicit a strong antibody response, but it also stimulates an antiviral (IFNγ) response among CD4+, CD8+, and CD4+CD8+ memory T cells along with NK cells and γδT cells.

## 2. Results

### 2.1. Vaccine Efficacy

Survival analysis reveals that vaccination with ASFV-G-ΔI177L is 100% effective at preventing death from virulent ASFV-G challenge at 28 days post-inoculation (dpi) (Figure 1A, *p*-value = 0.0011), with a median survival of placebo swine of 6 days post-challenge (dpc). Swine vaccinated with ASFV-G-ΔI177L become detectably viremic around 4–7 dpi, with all vaccinated swine remaining viremic by 28 dpi (Figure 1B), however, as previously described [7,8], the presence of attenuated vaccine strain virus in blood did not induce any detectable clinical signs, like elevated temperature (Figure 1C), or other signs like lethargy or anorexia (data not shown). Following challenge with wild type ASFV-G, viremia titers of placebo control animals increased over the limit of detection by 4–7 dpc, while the remanent viremia in the vaccinated group continued decreasing over time. The slope of the average serum virus titer among ASFV-G-ΔI177L-vaccinated swine was found to be significantly different than that of the control swine following challenge (*p*-value < 0.0001, Best fit slope ± standard error: 0.7533 ± 0.1265 vs. −0.1362 ± 0.02800 for Mock vs. ASFV-G-ΔI177L-vaccinated swine). Similarly, rectal temperatures of ASFV-G-ΔI177L-vaccinated swine, ranging between 39–40 °C from the time of inoculation on, did not vary much over the post-inoculation or post-challenge time points, while those of control swine spiked sharply from 0–7 dpc (Figure 1C). The slope of the average rectal temperature among ASFV-G-ΔI177L-vaccinated swine was found to be significantly different than that of the control swine following challenge (*p*-value < 0.0001, Best fit slope ± standard error: 0.4238 ± 0.08439 vs. 0.02461 ± 0.01627 for Mock vs. ASFV-G-ΔI177L-vaccinated swine).

### 2.2. Anti-ASFV Serum Antibodies

Serum antibodies against ASFV-G began to rise above the limit of detection around 7–11 dpi among ASFV-G-ΔI177L-vaccinated swine (Figure 1D) and remained high from about 14 dpi through the end of the challenge period. Anti-ASFV-G antibodies among control swine following challenge have been measured previously by this group but have never risen above the limit of detection prior to the humane endpoint, and thus, were not assessed in the present study.

### 2.3. Serum Cytokines

During ASF, infected monocyte/macrophages, the major targets of ASFV, produce cytokines that are related to mechanisms involved in disease pathogenesis [18,19]. More recently, studies have focused on the characterization of the systemic cytokine profile during infection with highly virulent ASFV strains [20]. Additionally, it has been described that ASFV strains of diverse virulence can trigger different patterns of soluble mediators (cytokines and chemokines) including IFN subtypes, IL-1β, IL-12p40 and TNFα, as compared to their virulent counterparts [21,22,23]. Analyzing the cytokine profile of ASFV-G-ΔI177L-vaccinated animals before and after challenge along with mock animals could help illuminate the mechanisms of induced protection.

#### 2.3.1. Serum Cytokines following Inoculation

Analysis of a variety of pro-inflammatory cytokines was performed at different times points in serum of vaccinated swine and compared with mock animals (Figure 2). Following vaccination, no significant differences were noted in serum cytokine levels of TNFα (Figure 2A), IFNα (Figure 2B), IL-1β (Figure 2C), IL-6 (Figure 2E), IL-8 or (Figure 2F). However, following ASFV-G-ΔI177L inoculation, serum concentration of IL-1Ra (Figure 2D) increased significantly over treatment-matched 0 dpi control at 4 (*p*-value = 0.0062) and 7 dpi (*p*-value = 0.0463) among ASFV-G-ΔI177L-vaccinated swine and their 0 dpi baseline controls, while by 21 dpi, the serum IL-1Ra concentration among mock-inoculated swine had decreased significantly compared to baseline (*p*-value = 0.0343). At 7 dpi the serum concentration of this analyte was also significantly higher among ASFV-G-ΔI177L-vaccinated swine compared to time-matched controls (*p*-value = 0.0336). Serum concentrations of IL-12p40 were also significantly elevated over baseline at 14 and 21 dpi (*p*-value = 0.0192 and *p*-value = 0.0404 respectively, Figure 2G) among ASFV-G-ΔI177L-vaccinated swine, and trended higher than their time-matched placebo controls at 21 dpi (*p*-value = 0.0534).

#### 2.3.2. Serum Cytokines following Challenge

Following challenge with ASFV-G, no significant differences were observed between treatments or over time in serum concentrations of TNFα (Figure 2H), IL-1β (Figure 2J) or IL-12p40 (Figure 2N). Conversely, serum cytokine levels of IFNα (*p*-value = 0.0351, Figure 2I), IL-1Ra (*p*-value = 0.0414, Figure 2K) and IL-8 (*p*-value = 0.0110, Figure 2M) were significantly higher at 7 dpi among mock-vaccinated swine compared to their 0 dpc baseline, while serum concentration of IL-12p40 trended higher among mock swine at this timepoint compared to their 0 dpc baseline (*p*-value = 0.0698). On the other hand, among ASFV-G-ΔI177L-vaccinated swine, concentrations of IFNα (*p*-value = 0.0043) and IL-6 (*p*-value = 0.0051, Figure 2L) were significantly lower than their treatment-matched baseline at 7 dpc. The serum concentration of IL-1Ra was significantly higher among mock-vaccinated swine compared to ASFV-G-ΔI177L-vaccinated swine (*p*-value = 0.0008) at 7 dpc, while IFNα (*p*-value = 0.0541) and IL-12 (*p*-value = 0.0805) trended higher among mock-vaccinated swine at 7 dpc than ASFV-G-ΔI177L-vaccinated swine.

### 2.4. Cellular Immune Response Landscape after ASFV-G-ΔI177L Inoculation

There is evidence that T cells and other leukocytes play a role in mounting a strong immune response to ASFV. A study by Oura et al. found that depletion of CD8+ T cells following exposure to a nonvirulent strain of ASFV resulted in a loss of protection against later challenge with a homologous strain [15]. As reviewed by Schäfer et al., γδT cells and NK cells are also believed to play a role in both antigen presentation and antiviral cytotoxicity, while Treg populations show modulation in response to ASFV [24].

#### 2.4.1. Memory T and NK Cells

In order to assess Memory T and natural killer (NK) cell function, a multicolor flow panel was employed (antibodies and fluorophore list can be found in Appendix A) and was gated as shown in Figure 3A, with gates drawn according to fluorescence minus one (FMO). PBMCs were stimulated ex vivo with ASFV-G at MOI 0.5 and all statistics reported are on the difference in either population percentages or median fluorescence intensity (MFI) between the well stimulated with ASFV-G and the unstimulated cells. Memory T cells were identified as those single-positive CD8 or CD4 T or double positive CD4+CD8+ cells that positively expressed CD44 and were negative for CD25 and CD27. Central memory cells (CD44+CD25−CD27−CD62L+CCR7+) were distinguished from effector memory cells (CD44+CD25−CD27−CD62L−CCR7−) by assessing CD62L and CCR7 expression. Populations could then be assessed for IFNγ expression. Confirmatory gating of T cell subsets (CD4+CD8−, CD4−CD8+ and CD4+CD8+) demonstrates that the vast majority of IFNγ positive cells upon ex vivo restimulation of PBMCs from vaccinated animals are CD44+ (data not shown). Following inoculation with ASFV-G-ΔI177L, many significant changes could be observed in the memory T cell compartment (Figure 3B). Among CD8+ T cells, central memory cells decreased significantly by 28 dpi compared to baseline (*p*-value < 0.0001) and were significantly lower than the placebo controls at that time (*p*-value < 0.0001) (Figure 4A), while effector memory increased significantly by 28 dpi (*p*-value < 0.0001), being significantly elevated above placebo controls at this time point (*p*-value < 0.0001) (Figure 4B). Upon ex vivo stimulation, the percentage of IFNγ+ cells also increased significantly in both CD8+ central (T_cm_) and effector (T_em_) memory cells (Figure 4C,D). At 28 dpi (*p*-value = 0.0004), the percentage of CD8+ T_cm_ positive for IFNγ was increased compared to baseline, while by 28 dpi it was also elevated above the placebo control group for that time point (*p*-value = 0.0040). The percentage of IFNγ+ CD8+ T_em_ were also significantly increased at 14 and 28 dpi (*p*-value < 0.0001 at both time points) over baseline as well as their placebo controls (*p*-value = 0.0001). MFI of IFNγ-PE was also assessed for IFNγ+ memory T cells and was found to be increased in CD8+ T_cm_ among ASFV-G-ΔI177L (*p*-value < 0.0174) at 14 dpi and 28 dpi (*p*-value < 0.0001) compared to their respective baseline levels, while a significant difference between ASFV-G-ΔI177L swine and their placebo controls was observed at 28 dpi (*p*-value = 0.0011) (Appendix A). When assessing CD8+ T_em_, IFNγ-PE MFI was significantly increased compared to baseline at 14 dpi (*p*-value = 0.0001) among ASFV-G-ΔI177L-vaccinated swine, as well as compared to their time-matched controls (*p*-value = 0.0409) (Appendix A). MFI of CD62L, a lymph node-homing adhesion protein, was also more likely to be downregulated in ex vivo stimulated CD8+ memory T cells. While among CD8+ T_cm_, CD62L MFI was not significantly decreased at any analyzed time point (Appendix A), in CD8+ T_em_, CD62L expression was significantly downregulated compared to baseline at 14 (*p*-value = 0.0003) and 28 dpi (*p*-value = 0.0011) among ASFV-G-ΔI177L swine, as well as compared to their time-matched controls (*p*-value = 0.0096 and *p*-value = 0.0014, respectively) (Appendix A). For all T cell subsets analyzed post-inoculation, no differences were seen in CD62L expression in unstimulated wells (Figure 3B, data not shown).

Among CD4+ T cells, T_cm_ decreased significantly by 28 dpi in vaccinated animals compared to baseline (*p*-value < 0.0001) as well as placebo controls (*p*-value < 0.0001), while there was no change in the percentage of this population among placebo controls over this time (Figure 4E). CD4+ T_em_, in contrast, increased among ASFV-G-ΔI177L swine by 28 dpi (*p*-values < 0.0001) compared to baseline, being significantly higher than that of placebo swine at the same time points (*p*-value < 0.0001) (Figure 4F). Upon ex vivo stimulation, no significant differences were seen between treatment groups or over time in the percentage of CD4+ T_cm_ that were also positive for IFNγ (Figure 4G), while at 14 dpi, the percentage of CD4+ T_em_ positive for IFNγ was significantly higher among ASFV-G-ΔI177L swine both compared to their baseline as well as the time-matched placebo controls (*p*-value < 0.0001 and *p*-value = 0.0004, respectively) (Figure 4H). The MFI of IFNγ expression in CD4+ T_cm_ was significantly higher at 28 dpi in the ASFV-G-ΔI177L swine versus baseline (*p*-value = 0.0275), however, no significant differences were seen between treatment groups at any time point (Appendix A). Among CD4+ T_em_, there was a significant increase in IFNγ MFI of ASFV-G-ΔI177L swine at only 14 dpi compared both to baseline as well as the placebo controls (*p*-values < 0.0001 and *p*-value = 0.0016, respectively) (Appendix A). The MFI of CD62L expression of CD4+ T_cm_ decreased at 28 dpi compared to baseline (*p*-value = 0.0352), though no significant differences were noted between treatment groups (Appendix A). Finally, CD62L MFI in CD4+ T_em_ was not significantly altered either by treatment or time post-inoculation (Appendix A).

Among CD4+CD8+ memory T cells, T_cm_ decreased significantly by 28 dpi in vaccinated animals compared to baseline (*p*-value < 0.0001) as well as placebo controls (*p*-value < 0.0001) (Figure 4I). CD4+CD8+ T_em_, in contrast, increased among ASFV-G-ΔI177L swine by 28 dpi compared to both baseline and the placebo control group (*p*-value < 0.0001) (Figure 4J). Upon ex vivo stimulation, CD4+CD8+ T_cm_ positive for IFNγ increased significantly at 14 dpi among ASFV-G-ΔI177L swine compared to their baseline (*p*-value = 0.0020), though not against the time-matched placebo controls (Figure 4K). Meanwhile CD4+CD8+ T_em_, responded more strongly, with ASFV-G-ΔI177L swine achieving a highly significant increase over both baseline, as well as the placebo control group at both 14 and 28 dpi (*p*-values < 0.0001) (Figure 4L). The MFI of IFNγ expression in CD4+CD8+ T_cm_ was significantly higher in placebo control swine at 28 dpi compared to baseline (*p*-value = 0.03522), while no other differences were noted in this cell type (Appendix A). Among CD4+CD8+ T_em_, the MFI of IFNγ expression in ASFV-G-ΔI177L swine was significantly lower than the time-matched placebo control group at 7 dpi (*p*-value = 0.0091), while IFNγ expression MFI was significantly elevated over baseline at 14 dpi among ASFV-G-ΔI177L swine (*p*-value = 0.0153) (Appendix A). Analysis of CD62L expression revealed no differences in the CD4+CD8+ T_cm_ population (Appendix A), while among T_em_, in contrast, ASFV-G-ΔI177L swine demonstrated significant downregulation of this marker upon ex vivo stimulation compared to baseline at 14 and 28 dpi (*p*-value < 0.0001 and *p*-value = 0.0197, respectively), while at these timepoints, CD62L MFI was also significantly downregulated compared to placebo control swine (*p*-value = 0.0026 and *p*-value = 0.0289, respectively) (Appendix A).

We were also interested in determining the effect of vaccination with this live attenuated strain on NK cells, due to the strong antiviral activity that they exhibit. We observed a significant difference in the percentage of NK cells (CD3−CD4−CD8+) [25] between treatment groups at 7 dpi, with placebo controls possessing significantly more NK cells than the ASFV-G-ΔI177L swine (*p*-value = 0.0162), however, this was the only significant difference noted in this population post-inoculation (data not shown). The percentage of NK cells positive for IFNγ increased significantly amongst ASFV-G-ΔI177L swine at 14 and 28 dpi when compared both to baseline as well as the time-matched placebo-controls (*p*-values < 0.0001) (Figure 4M). No significant differences were seen, however, in the MFI of IFNγ between populations or over time (data not shown).

#### 2.4.2. γδT Cells

In order to assess changes in percentages of regulatory as well as γδT cells, a separate multicolor flow panel was employed (Appendix A). The gating strategy for this panel was based upon FMOs (Figure 5A). γδT cells are a highly populous circulating subset of T cells in swine and other artiodactyls, though their role in mediating ASFV infection and immunity is not well-understood. Several changes were seen in the representation of this T cell subset as well as its IFNγ expression profile following vaccination with ASFV-G-ΔI177L. The percentage of γδT cells (γδTCR+) as a proportion of the parent population (CD3+CD4−CD8−) decreased significantly compared to baseline among placebo control swine at 4 (*p*-value = 0.0489) and 28 dpi (*p*-value < 0.0001) and among ASFV-G-ΔI177L swine at 28 dpi (*p*-value = 0.0003), while the reduction among ASFV-G-ΔI177L compared to time-matched control was only significant at 14 dpi (*p*-value = 0.0072) (Figure 5B). The percentage of γδT cells positive for IFNγ expression increased significantly over baseline among ASFV-G-ΔI177L swine at 7 (*p*-value = 0.0048), 14 (*p*-value < 0.0001) and 28 dpi (*p*-value < 0.0001), while this increase was significantly more than the time-matched control swine at 14 (*p*-value < 0.0001) and 28 dpi (*p*-value < 0.0001) (Figure 5C). The MFI of IFNγ expression in this population among ASFV-G-ΔI177L swine was significantly greater than baseline at 14 (*p*-value = 0.0003) and 28 dpi (*p*-value = 0.0075), while the increase over time-matched controls was also significant at 14 (*p*-value = 0.0014) and 28 dpi (*p*-value = 0.0054) (Appendix A).

#### 2.4.3. Regulatory T Cells

Regulatory T cells (T_reg_) play an important role in mediating the intensity of the immunological response to infectious agents. Among ASFV-G-ΔI177L swine, T_reg_ (CD3+CD4+CD8−CD25+FOXP3+) were significantly increased over both baseline (*p*-value < 0.0001) as well as time-matched control swine at 7 dpi (*p*-value = 0.0015) (Figure 5D). There were no significant differences in the change in percentage of Treg positive for IFNγ (Figure 5E), while the MFI of IFNγ expression in this population was significantly greater than baseline among ASFV-G-ΔI177L swine at 14 dpi (*p*-value = 0.002) (Appendix A).

#### 2.4.4. Innate Immune Cell Populations

Following inoculation, PBMCs were assessed for various myeloid cell populations using another multicolor flow cytometry panel (Appendix A). Populations assessed include CD3−CD14−CD172a+SLA−II+ (Dendritic cell-like [DCs-like]), CD3−CD14−CD172a+SLA−II+, CD4− (conventional DC-like [cDC-like]) and CD3−CD14−CD172a+SLA−II+, CD4+ (plasmacytoid DC-like [pDC-like]), along with CD3−CD14+ (monocytes) [25]. For the most part, changes over time and between treatment groups of cell populations and MFI of various markers were stochastic and not significant (results summarized in Appendix A).

### 2.5. Cellular Immune Response Landscape after Challenge with Homologous ASFV-G

As mentioned before, there is abundant evidence of induction of a cellular immune response after inoculation with attenuated ASFV strains [24], while animals infected with virulent ASFV strains died before memory cells could have formed [26]. Understanding progression of immune response of immunized pigs after challenge may help understand mechanism of attenuation and ultimately improve vaccine design.

#### 2.5.1. Memory T and NK Cells

Following challenge at 28 dpi, all placebo control swine were euthanized at either 6 or 7 dpc, when they met humane endpoint criteria. Over the course of infection, no significant changes were observed in memory T cell populations or intensity of IFNγ or CD62L expression among placebo challenged animals as compared to status of these animals at 0 dpc (Figure 6 and Appendix A). However, between challenge and euthanasia, many significant differences were observed between ASFV-G-ΔI177L vaccinated swine and both placebo controls as well as baseline in memory T cell populations (Figure 6). At 0 dpc (*p*-value = 0.0016) the percentage of CD8+ T_cm_ was significantly reduced among ASFV-G-ΔI177L swine compared to placebo-vaccinated swine, while at 7 dpc (*p*-value = 0.0095) and 14 dpc (*p*-value = 0.0137), the reduction in T_cm_ among ASFV-G-ΔI177L swine was significantly reversed compared to 0 dpc (Figure 6A). CD8+ T_em_ were significantly greater among ASFV-G-ΔI177L swine compared to placebo control swine at 0 (*p*-value = 0.0006) and 4 dpc (*p*-value = 0.0048) (Figure 6B). The percentage of CD8+ T_cm_ positive for IFNγ increased following challenge, being significantly higher than baseline among ASFV-G-ΔI177L swine at 4 (*p*-value < 0.0001) and 7 (*p*-value = 0.0001) dpc and compared to placebo controls at 4 (*p*-value < 0.0001) and 7 dpc (*p*-value = 0.0001 (Figure 6C). The percentage of CD8+ T_em_ positive for IFNγ also increased over time following challenge, being significantly higher than baseline among ASFV-G-ΔI177L swine at 7 (*p*-value = 0.0013) and 14 dpc (*p*-value = 0.0015), and also compared to placebo controls at 0 (*p*-value = 0.0021), 4 (*p*-value = 0.0029) and 7 dpc (*p*-value < 0.0001) (Figure 6D). No significant changes were observed in the MFI of IFNγ-PE among CD8+ T_cm_ over time, though a small effect was noted between treatment groups at 4 dpc (*p*-value = 0.0259) (Appendix A). Among CD8+ T_em_ MFI of IFNγ-PE was not significantly different across treatment groups or over time, though among ASFV-G-ΔI177L swine at 7 dpc (*p*-value = 0.0740) there was a trending increase over placebo control swine (Appendix A). Expression levels of CD62L on CD8+ T_cm_ were also significantly reduced among ASFV-G-ΔI177L swine compared to placebo controls at 7 dpc (*p*-value = 0.0415) (Appendix A). Among ASFV-G-ΔI177L swine, expression of CD62L on CD8+ T_em_ was downregulated at 7 (*p*-value = 0.0001) and 14 dpc (*p*-value = 0.0069) compared to baseline and placebo controls at 7 dpc (*p*-value = 0.0415) (Appendix A).

A variety of changes were also observed in CD4+ memory T cell subsets following challenge. At 0 and 4 dpc the percentage of CD4+ T_cm_ was significantly reduced upon ex vivo stimulation when compared to the corresponding placebo swine (*p*-value < 0.0001 and *p*-value = 0.0368, respectively), while at 7 dpc, the reduction in the percentage of this population among ASFV-G-ΔI177L vaccinated swine was significantly reversed compared to baseline (*p*-value = 0.459) (Figure 6E). CD4+ T_em_ were significantly elevated among ASFV-G-ΔI177L vaccinated swine compared to placebo control swine at 0 (*p*-value = 0.0001), 4 (*p*-value = 0.0002) and 7 dpc (*p*-value = 0.0004) (Figure 6F). The percentage of CD4+ T_cm_ positive for IFNγ was increased among ASFV-G-ΔI177L-vaccinated swine compared both to 0 dpc as well as the time-matched placebo control swine at 4dpc (*p*-values < 0.0001) (Figure 6G). No significant differences were noted in the percentage of CD4+ T_em_ positive for IFNγ (Figure 6H). Additionally, no significant differences were noted in the IFNγ MFI of CD4+ T_cm_ following challenge (Appendix A), while at 4 dpc IFNγ MFI in CD4+ T_em_ among ASFV-G-ΔI177L-vaccinated swine was significantly higher than placebo control swine (*p*-value = 0.0363), while at 7 dpc, IFNγ MFI among ASFV-G-ΔI177L-vaccinated swine was significantly higher than both baseline (*p*-value = 0.0017) as well as time-matched controls (*p*-value = 0.0345) (Appendix A). CD62L expression on CD4+ T_cm_ was significantly reduced among ASFV-G-ΔI177L vaccinated swine compared to both baseline (*p*-value = 0.0170) as well as placebo controls at 4 dpi (*p*-value = 0.0291) (Appendix A). However, no significant differences in CD62L MFI in CD4+ T_em_ were noted either over time or across treatment groups (Appendix A).

A number of significant changes were also noted following challenge in CD4+CD8+ double positive T cells, a population noted as playing a role in fighting viral infections such as ASFV in swine [25,27,28]. Among CD4+CD8+, the reduction in the percentage of T_cm_ following ex vivo stimulation was significantly reversed at 4 (*p*-value = 0.0031), 7 (*p*-value = 0.0009) and 14 dpc (*p*-value < 0.0001) among ASFV-G-ΔI177L swine compared to baseline, while this population was significantly more suppressed in ASFV-G-ΔI177L vaccinated swine compared to placebo controls at 0 dpc only (*p*-value < 0.0001) (Figure 6I). Among CD4+CD8+ T_em_, the increase in percentage of the population was significant in ASFV-G-ΔI177L treated swine at 0 (*p*-value = 0.0008) and 7 dpc (*p*-value = 0.0203) compared to placebo controls (Figure 6J). Following challenge and ex vivo stimulation in ASFV-G-ΔI177L vaccinated swine, the percentage of CD4+CD8+ T_cm_ positive for IFNγ was significantly higher at 4 and 7 dpc than both baseline (*p*-value = 0.0007 and *p*-value = 0.0002, respectively) as well as the placebo control swine (*p*-value = 0.0006 and *p*-value = 0.0020, respectively) (Figure 6K). The percentage of CD4+CD8+ T_em_ positive for IFNγ was significantly higher at 0 (*p*-value = 0.0015) 4 (*p*-value = 0.0009) and 7 dpc (*p*-value = 0.0024) than placebo control swine (Figure 6L). Analysis of IFNγ-PE MFI following ex vivo stimulation in CD4+CD8+ T_cm_ revealed only a small significant increase in this marker’s expression at 7 dpc (*p*-value = 0.0256) compared to baseline in ASFV-G-ΔI177L vaccinated swine (Appendix A). Among CD4+CD8+ T_em_, a significant increase in IFNγ-PE MFI was noted in both the placebo control (*p*-value = 0.0012) and ASFV-G-ΔI177L treatment groups (*p*-value = 0.0031) compared to their respective baselines (Appendix A). Investigation of the change in expression of CD62L following ex vivo stimulation revealed no significant differences either across time or treatment groups in either CD4+CD8+ T_cm_ (Appendix A) or T_em_ (Appendix A).

In the case of NK cells, following challenge, the proportion of these cells among ASFV-G-ΔI177L treated swine was significantly higher at 7 dpc compared to baseline (*p*-value = 0.0473), while at the same time point, the percentage of NK cells was significantly lower than baseline among placebo control swine (*p*-value = 0.0017) (data not shown). However, upon ex vivo stimulation, the percentage of NK cells positive for IFNγ was significantly higher among ASFV-G-ΔI177L vaccinated swine compared to placebo controls at 0 (*p*-value = 0.0175), 4 (*p*-value < 0.0001) and 7 dpc (*p*-value = 0.0008) and higher than baseline among ASFV-G-ΔI177L treated swine at 7 dpc (*p*-value < 0.0001) and 14 dpc (*p*-value = 0.0058) (Figure 6I). MFI of IFNγ in NK cells was significantly downregulated among placebo control swine at 7 dpc compared to baseline (*p*-value = 0.0309), while the MFI was significantly higher among ASFV-G-ΔI177L swine compared to placebo controls at this timepoint (*p*-value = 0.0433) (data not shown).

#### 2.5.2. γδT Cells

γδT cell populations continued to change after challenge. Among mock-vaccinated swine, there was a significant decrease in γδT cells at 4 (*p*-value = 0.0203) and 7 dpc (*p*-value = 0.0012) compared to baseline within ex vivo unstimulated cells, while there was a significant increase in γδT cells by 7 dpc (*p*-value = 0.0032) compared to baseline among ASFV-G-ΔI177L swine (Figure 7A). No significant differences were noted within treatment groups over time in the percentage of γδT cells positive for IFNγ staining following challenge, although between groups, there were significantly more IFNγ+ γδT cells among the ASFV-G-ΔI177L vaccinated animals compared to control swine at 0 dpc (*p*-value = 0.0057) and a trending increase at 7 dpc (*p*-value = 0.0596) (Figure 7B). The MFI of IFNγ expression in γδT cells among ASFV-G-ΔI177L treated swine was significantly greater than that of control swine at 0 dpc (*p*-value = 0.004), while it decreased significantly compared to baseline by 14 dpc (*p*-value = 0.003) (Appendix A).

#### 2.5.3. Regulatory T Cells

Regarding variation in the percentage of T_reg_ after challenge, no significant differences were observed over time nor between treatment groups (*p*-values > 0.05) (Figure 7C). The percentage of T_reg_ positive for IFNγ decreased upon ex vivo stimulation in mock-vaccinated swine at 4 (*p*-value = 0.0136) and 7 dpc (*p*-value = 0.0213) compared to baseline, while no significant changes were seen between treatment groups nor over time in the ASFV-G-ΔI177L swine (Figure 7D). Among ASFV-G-ΔI177L swine, IFNγ MFI of IFNγ+ T_reg_ increased significantly at 7 dpc (*p*-value = 0.0293) compared to baseline, though no other significant changes in MFI were noted in IFNγ+ T_reg_ (Appendix A).

#### 2.5.4. Innate Immune Cell Populations

Following challenge, PBMCs were again assessed for various myeloid cell populations. Similar to the post-inoculation time points, differences over time and between treatment groups were generally non-significant. Results have been summarized in Appendix A.

## 3. Discussion

The current study provides a detailed analysis of both the humoral and cellular immune response landscape following vaccination with a highly effective live attenuated ASF vaccine, along with the immune landscape following challenge with its wild-type homologue. Development of a highly effective vaccine against ASFV has long been an elusive goal, due in no small part to: a lack of consensus on what defines a protective immune response; the difficulties involved with studying a virus that requires a high level of biocontainment; and the challenges of working with a virus without any closely-related viral relatives except for a recently discovered putative relative, which infects abalone and is far less well-understood even than ASFV [29]. Our study confirms past studies out of our lab showing that this novel vaccine candidate elicits a strong humoral immune response within the first 7–11 days post vaccination [7,8]. In fact, serum anti-ASFV IgG titer among ASFV-G-ΔI177L-vaccinated swine in the current study plateaued by 11 dpi. In a related study with this vaccine candidate, Tran et al. observed 50% mortality among Vietnamese swine and ~30% mortality in European swine challenged with a locally circulating Vietnamese strain of ASFV-G at 14 dpi, while survival in both pig strains rose to 100% when challenged just 7 days later at 21 dpi, suggesting that high antibody titers may not be sufficient on their own to protect against wild-type challenge [9]. A 2015 study assessing the protective efficacy of an ASFV-G-Δ9GL vaccine candidate found that the numbers of IFNγ-producing cells, which rose between 8–28 dpi, were well-correlated with protection from homologous challenge in their vaccinated group [30]. Conversely, a 2017 study by the same group of a double-gene knockout vaccine candidate ASFV-G-Δ9GL/ΔUK found that survival of vaccinated swine was 100% when challenged at 14 dpi and did not seem to be correlated with T cell immunity [14]. However, the method used to assess this endpoint in both studies was an IFNγ ELISpot, which does not take cell type or memory polarization into account. In a 2016 study by Carlson et al. using an ASFV-Pretoriuskop/96/4-Δ9GL single gene deletion vaccine candidate, the authors found that survival increased in a stepwise fashion from 40% to 80% when vaccinated swine were later challenged at 7, 10, 14, 21 or 28 dpi with the parental strain [13]. The authors found no clear correlation at any challenge time point between survival and IFNγ-producing cells, as measured by IFNγ ELISpot, nor was there any strong correlation between survival and any of the measured parameters at the 7, 10 and 14 dpi challenge groups, while at the later challenge time points, 21 and 28 dpi, ASFV-specific antibodies became a more reliable correlate of protection. In the present study, while the proportions of both CD8+, CD4+ and CD4+CD8+ T_mem_ expressing IFNγ begin to rise to the level of significance by 14 dpi, in terms of T_mem_ orientation to either central or effector phenotype upon ex vivo restimulation, this response does not become significant in vaccinated swine until 28 dpi. Interestingly, we see a sharp spike in IFNγ MFI at 14 dpi among CD8+ T_cm_ as well as CD8+, CD4+ and CD4+CD8+ T_em_. By 28 dpi, IFNγ MFI has returned to a non-significant level in all T_em_ populations, while remaining elevated, in CD4+ and CD8+ T_cm_. Meanwhile, an anomalously high IFNγ-PE MFI among CD4+ T_cm_ from a control pig at 14 dpi along with another high IFNγ-PE MFI among CD4+CD8+ T_cm_ from two control swine at 28 dpi reflect noise in the data—the number of events falling within the IFNγ+ gate used to determine these 3 MFI was either 1 or 2 in each case, an indication of just how unresponsive T cells from control pigs were to ex vivo stimulation with ASFV-G. Taken together, this may suggest that it is not enough to have IFNγ-producing, ASFV-G-reactive T cells by 14 dpi, but that proper orientation of those T cells as either effector or central memory may play a role in protective immunity. It is important to mention here that even though vaccinated animals showed detectable levels of viremia peaking at between 7 and 14 days post-inoculation, viremia continued to decrease even after challenge, indicating that challenge virus was effectively neutralized at the inoculation site by the primed immune cells and/or circulating neutralizing antibodies. A study by Iyer et al. in 2009 found that a highly effective live attenuated recombinant vaccine against West Nile Virus (a virus that infects monocytes in humans) was also correlated with reduced CD62L expression in IFNγ+ CD8+ T cells prior to challenge [31]. A 2005 study of a live attenuated human immunodeficiency virus (HIV) vaccine candidate constructed by incorporating HIV gene p160 into the vaccinia virus sought to establish a more thorough phenotype of memory T cells associated with long-term proliferative capacity [32]. The authors found that while HIV-tetramer+ CD8+ T cells underwent an abrupt and transient downregulation of CD62L expression in the first several days following inoculation, expression levels of this marker continued to rise out to 250 dpi and were not correlated with the ability of these cells to proliferate or produce IFNγ or TNFα in response to in vitro stimulation, similar to the current study’s finding that IFNγ production capacity does not correlate with surface CD62L expression. Other viruses such as Ebola virus [33], HIV [34,35], and others, encode proteins that inhibit and modulate CD62L expression and ectodomain shedding as part of their host immune evasion strategy. It has been observed that the shedding of CD62L is necessary for CD8+ T cells to gain lytic activity, as assessed by CD107a expression [36]. Future studies of ASF vaccine candidates ought to measure the full polarization of memory type T cells in terms of both CD62L and CCR7 expression, and should consider not only IFNγ expression, but also CD107a, perforin, granzyme B and granulysin expression, as well as the direct cytolytic capacity of these memory cells over time.

In the present study, a strong Type II IFN response was noted in NK cells by 14 dpi among ASFV-G-ΔI177L-vaccinated swine, and this response remained strong after challenge, suggesting that perhaps these cells play a role in developing antiviral immunity and in fighting infection. A positive correlation between NK cells and protection from ASFV has also been documented in a previous study of the attenuated strain ASFV/NH/P68 [16]. Conversely, a study using the virulent Malta 78 strain of ASFV demonstrated suppression of NK cell activity between 3 and 6 dpi [37], which is in agreement with the lack of NK cell IFNγ activity noted post-challenge in our mock-vaccinated swine. These results suggest that the ability of NK cells to participate in ASFV immunity is dependent upon the virulence of the strain used. In a study of NK cell activity in vitro using both a non-virulent and virulent strain, Martins and Leitão found that while the non-virulent NK/P68 ASFV strain stimulated NK cell activity, exposure of NK cells to the virulent Lisbon 60 strain suppressed their activity [38]. Taken together, all of these findings concord well with the results of the present study. On the other hand, it is surprising to observe such a strong NK cell response upon re-stimulation from 14 dpi onwards. It is possible, that NK cells from ASFV-G-ΔI177L-vaccinated swine are undergoing an increased non-specific response to a secondary stimulation known as “trained immunity”, as previously described for cytomegalovirus (another double stranded DNA virus) [39,40]. Future research should further explore this understudied innate immune cell and its role in vaccine-induced ASFV immunity.

While several other studies have measured changes in percentages of γδT cells and T_reg_, to the best of our knowledge, the current study is the first to assess antigen-specific responsiveness of these cell subsets via IFNγ production by flow cytometric analysis. The proportion of γδT cells that were IFNγ+ increased significantly by 7 dpi in vaccinated swine, producing prolific amounts of the cytokine at what is a relatively early time point compared to αβT cells and NK cells. Hühr et al. also observed a significant decrease in the percentage of γδT cells in the blood by 7 dpc in response to the highly virulent strain ASFV-Armenia08 in commercial swine [27], while a study by the same group observed a significant, yet transient, increase in this population at 5 dpi in commercial swine in response to infection with the milder ASFV-Estonia2014 [28]. However, neither study measured Type II IFN response in these cells. On 7 dpi, there was a simultaneous spike in the relative abundance of T_reg_ among vaccinated swine following ex vivo stimulation with ASFV-G. A similar spike in T_reg_ was also observed in the 2020 paper by Hühr et al. in the blood and spleen of both domestic swine and wild boars, as well as the gastrohepatic lymph node of commercial swine at 7 dpc with the virulent ASFV-Armenia08 strain. A more recent paper by the same group revealed a similar increase in T_reg_ by 10 dpc in the lungs of commercial swine and the spleen and lungs of wild boars following challenge with a strain less virulent to commercial swine ASFV-Estonia2014 [28]. Interestingly, we see no commensurate spike in T_reg_ following challenge in our mock-vaccinated swine, again pointing to the potential importance of strain virulence in the magnitude and temporality of the cellular immune response to infection. Possibly this early spike in T_reg_ cells functions to rein in and control the αβT cell, γδT cell and NK cell pro-inflammatory response against the virus. While there is not much in the virology literature regarding IFNγ+ T_reg_, the spike in induction of this cytokine at 14 dpi may be related to control and orientation of the virus response. In cancer literature, IFNγ induction and IL-10 suppression in T_reg_ appears to be an early requirement for T_reg_ to properly mediate an anti-tumor environment, clear more tolerant T_reg_ from their surroundings and boost anti-tumor immunity [41,42]. In ASFV literature, detectable circulating levels of IL-10 are associated with a derailed immune response to ultimately fatal ASFV infection [24]. Given this information, measuring intracellular and circulating IFNγ and IL-10 may be useful in future studies of ASF vaccine candidates to assess the appropriateness of the immune response.

Even though few changes were seen in either DC-like cells or monocyte numbers as a percentage of the parent population following ex vivo stimulation with ASFV-G, only few changes were seen with relative consistency in terms of MFI of the surface markers SLA-II and CD14. SLA-II, the swine MHC-II surface protein, was upregulated at every timepoint following ASFV-G-ΔI177L-inoculation among vaccinated swine (4–28 dpi), as well as at 4, 7 and 14 dpi in control swine (Appendix A). While the cause of the upregulation in control swine is unclear, the result of an upregulation of MHC-II in ASFV-G-ΔI177L-inoculated swine is increased antigen presentation to and activation of CD4+ T cells. At 14 and 28 dpi, we also observed a significant downregulation in the expression of CD14, an LPS receptor, compared to baseline among ASFV-G-ΔI177L-vaccinated swine, along with a significantly lower expression at 14 dpi compared to mock-vaccinated swine. A recent in vitro study found that infection of macrophages and monocytes in culture with both a tissue culture adapted avirulent strain ASFV-BA71V as well as a virulent Sardinian strain ASFV-22653/14 resulted in significantly reduced CD16 expression, consistent with our findings, and a lack of change in MHC-II expression on these cells, inconsistent with our findings [43]. In another in vitro study of adherent porcine bone marrow cells, infection with virulent ASFV-Benin97/1 was again correlated with downregulated surface expression of CD16 and unaltered MHC-II expression [44]. In a follow-up study, Franzoni et al. found that monocyte-derived macrophages (moMφ) infected in vitro with virulent ASFV-22653/14 had a reduced capacity to release IL-6, IL-12 and TNFα upon classical stimulation or stimulation with a TLR2 agonist [21]. While this effect was partially abrogated during infection with the attenuated strain ASFV-NH/P68, production of these cytokines was still impaired compared to mock-infected cells. Taken together, these results suggest that ASFV is replicating surreptitiously inside its target cells and evading immune surveillance early in infection. While the upregulation of SLA-II observed in the current study is encouraging, the downregulation of CD16 may contribute to a slower cell-mediated immune response to the vaccine strain and thus delay the onset of full, protective immunity. Future studies of genetically attenuated vaccine candidate strains should incorporate in vitro assessment of monocyte, moMφ, and Mφ disfunction into their suite of immunogenicity testing to ensure optimal and timely effector cell priming and activation downstream of these target APCs.

Interestingly, while it has been observed that strong cytokine responses are necessary for eliciting an immune response against ASFV [24], this study found very little modulation of a variety of proinflammatory cytokines following immunization, possibly as a downstream result of the downregulation of CD16 observed in monocytes. While nonsignificant trends appear in the data such as a peak in serum TNFα at 7 dpi and a small elevation in IFNα among ASFV-G-ΔI177L at 4 dpi, the only significant increases in serum cytokines among vaccinated swine were at 4 and 7 dpi for IL-1Ra and closer to challenge, at 14 and 28 dpi for IL-12p40. While Type I IFNs and IL-12 are crucial co-stimulatory signals for promoting IFNγ production and cytotoxic capacity in NK and T cells [24], the timing and serum levels observed post-inoculation in the current study do not correlate well with the IFNγ responses we clearly see in our NK and T cells. For example, significantly elevated levels of IL-12p40 are observed coincident with elevations in intracellular IFNγ expression, rather than prior to. Curiously, while we observed strong Type II IFN responses upon ex vivo stimulation of NK and T cells, serum cytokine levels of IFNγ never rose above the limit of detection, despite using two separate commercially available detection kits. This finding, while curious, is similar to findings in a recent study by Wang et al. [20]. The Multigene Family (MGF) genes MGF360 and MGF505 play a role in downregulating Type I IFN release from infected monocytes and macrophages early in infection [23,45]. While studies of vaccine candidates involving deletions of the MGF genes demonstrate attenuation of the virus and protection against subsequent homologous challenge, IFN responses were either not measured [46] or were inconclusive [23]. While Reis and colleagues found that knocking out MGF360 and MGF505 from highly virulent ASFV-Benin resulted in increased mRNA levels of IFNβ in macrophages co-cultured with the virus, along with high serum concentrations of IFNγ in swine vaccinated with this candidate strain between 5 and 7 dpi, IFNγ ELISpot revealed low numbers of T and/or NK cells producing this cytokine at 46 dpi and only modest increases following challenge. In another recent study by Ran et al., the authors identified GC-enriched regions of the ASFV-NH/P68 genome as having a strong inverse correlation with IFNβ, IL-6 and TNFα expression, particularly the gene I267L, which they found inhibits the RIG-I pathway via interaction with Riplet [47]. Deletion of this gene yielded a vaccine candidate with 80% efficacy and significantly elevated serum IFNβ concentration at 5 dpc. In our study, we observed an increase in some of the analyzed cytokines directly correlated with disease in control swine following challenge. Conversely, vaccinated animals—none of which displayed clinical disease—generally showed no change or a slight decrease in serum concentrations of most cytokines following challenge.

The current study represents the first time such an exhaustive and cell-type-specific analysis of IFNγ production following ex vivo stimulation has been reported for either a wild-type or an attenuated strain of ASFV. While we observe that high antibody titers correlate with protection, we also observe strong correlation with: skewing of memory T cells away from a central memory phenotype and towards an effector memory phenotype; an upregulation of IFNγ production in CD8+ memory T cells, NK cells and γδT cells; and a downregulation of IFNγ production in CD4+ T_em_ cells. These correlates of protection were largely maintained in the post-challenge period for ASFV-G-ΔI177L-vaccinated swine, while control swine showed no signs of an adaptive immune response following challenge. It is worth mentioning that although it has been previously demonstrated that the levels of specific antibody titers induced by vaccination are the same regardless of vaccine dose used [8], studying the effect of vaccine dosage on the stimulation of cellular immune response deserves further research. While this single gene deletion LAV vaccine provided perfect protection and strong correlates of protection measurements at the 28 dpi challenge over this and several other studies [7,8,9], future studies should assess how durable this immunity is many months after inoculation. Future work should also explore the mechanistic underpinnings of the findings observed here. For example, future studies may consider focusing on: the involvement of monocytes and DCs on initial reaction to attenuated vaccine strains; how early antigen presenting cells responses affect downstream memory T cells stimulation; or what epigenetic and metabolic changes occur in NK cells following inoculation that result in the exuberant IFNγ response following restimulation, among others.

## 4. Materials and Methods

### 4.1. Virus and Cells

Primary swine macrophage cultures were used for growth and titration of the virus in 96-well plates as previously described [48].

Development of the ASFV-G-ΔI177L live attenuated vaccine candidate has also been described previously [8]. Briefly, a CRISPR gene editing system was used to partially delete the I177L gene from the genome of highly virulent ASFV-G, replacing it with an mCherry reporter under the ASFV p72 promoter.

Virus titration was performed on primary swine macrophages in 96-well plates, with viral dilutions being performed in macrophage media composed of RPMI 1640 Medium (Life Technologies, Grand Island, NY, USA) with 30% L929 supernatant and 20% fetal bovine serum (HI-FBS, Thermo Scientific, Waltham, MA, USA). Positive wells were identified by the presence of hemadsorption (HAD) [8], and the median HAD (HAD_50_) was determined by the Reed-Muench method [49].

Nino Vepkhvadze from the Laboratory of the Ministry of Agriculture (LMA) in Tbilisi, Republic of Georgia kindly provided the field isolate ASFV-G, which was used in the animal challenge.

### 4.2. Animals Studies

All animal experiments were performed under agricultural biosafety level 3 at Plum Island Animal Disease Center (PIADC) and were approved by the PIADC Institutional Animal Care and Use Committee of the U.S. Department of Agriculture and the U.S. Department of Homeland Security (Protocol #: 225.06-19-R). Crossbred Yorkshire swine of 40–50 lb were utilized to assess the efficacy and immunogenicity of the ASFV-G-ΔI177L vaccine against the parental ASFV-G strain. Groups of 10–12 swine were intramuscularly (i.m.) inoculated with 10^6^ HAD_50_ of ASFV-G-ΔI177L vaccine or a vehicle control (Dulbecco’s Modified Eagle Medium [DMEM], Life Technologies) at 0 dpi, monitored for 28 days and challenged i.m. with 10^2^ HAD_50_ of parental ASFV-G. While all swine were vaccinated or mock-vaccinated at 0 dpi, one group of 5–6 swine/treatment group was followed only till 28 dpi, while the remaining 5–6 swine/treatment group were then challenged and followed up to 20 dpc. For sampling purposes and following IACUC recommendations (including avoiding bleeding animals that have developed hematomas in previous bleedings), there are some time points that do not include all animals within the group, as indicated in each figure graph. Throughout the experiment, clinical signs (anorexia, depression, fever, purple skin discoloration, staggering gait, diarrhea, body temperature and cough) were recorded daily.

#### 4.2.1. Blood Processing

Blood samples were collected at 0, 4, 7, 14 and 28 dpi, then 4, 7 and 14 dpc in serum or heparinized Vacutainers (BD, Franklin Lakes, NJ, USA) to assess viremia, serum anti-ASFV antibodies and purified peripheral blood mononuclear cell (PBMC) populations. Serum tubes were allowed to coagulate for at least 30 min and centrifuged at 2100× *g* for 10 min at RT before aliquots were collected and stored at −70 °C until analysis. Blood from heparin tubes was transferred to Uni-Sep Maxi Plus 50 mL conical tubes and centrifuged at 800× *g* for 20 min with the brake turned off. PBMC layer was collected and washed with 1× Dulbecco’s phosphate-buffered saline (DPBS) before red blood cell lysis. Purified PBMCs were counted on a Vi-Cell Blu (Beckman Coulter, Brea, CA, USA) and either plated immediately for flow cytometry or cryopreserved in liquid nitrogen storage in fetal bovine serum (FBS) with 10% dimethyl sulphoxide (DMSO).

#### 4.2.2. Detection of Viremia

Virus titration of serum was performed as described above.

#### 4.2.3. Detection of Anti-ASFV Antibodies

Presence of virus-specific antibodies in the sera of inoculated swine was assessed with an in-house indirect ELISA, described elsewhere [13]. Briefly, the virus antigen was produced in Vero cells infected with a Vero-adapted ASFV strain. ELISA plates (Maxisorp, Nunc, St. Louis, MO, USA) were coated with infected or uninfected cell extract (1 μg/well). Plates were blocked with 10% skim milk (Merk, Kenilworth, NJ, USA) in PBS and 5% normal goat serum (Sigma, St. Louis, MO, USA). Each swine serum sample was tested at multiple dilutions against both infected and uninfected cell antigens. ASFV-specific antibodies in the swine sera were detected by a horseradish peroxidase-conjugated anti-swine IgG antibody (KPL, Gaithersburg, MD, USA) and SureBlue Reserve peroxidase substrate (KPL). Optical density was read at 630 nm (OD_630_) in an ELx808 plate reader (BioTek, Shoreline, WA, USA). Serum titers were expressed as the log_10_ of the highest dilution where the OD_630_ reading of the tested sera at least doubles the reading of the mock-infected sera.

#### 4.2.4. Serum Cytokine Analysis

Single analyte ELISA kits were used to determine the serum concentrations of the following cytokines: IFNα (Sigma, Cat#: RAB1131), IFNγ (Invitrogen, Cat#: KSC4021), IL-1β (R&D Systems, Cat#: PLB00B), IL-1Ra (R&D Systems, Cat#: DY780), IL-6 (R&D Systems, Cat#: P6000B), IL-8 (R&D Systems, Cat#: P8000), IL-12 (R&D Systems, Cat#: DY912), TNFα (R&D Systems, Cat#: PTA00). Serum was diluted 1:2 and all manufacturer’s protocols were followed. IL-1β samples with an optical density (O.D.) below that of background were converted to zeros for graphing and ones for log_10_ transformation and statistical analysis.

#### 4.2.5. Flow Cytometric Analysis

1 × 10^6^ PBMCs/well were plated in either duplicate (Myeloid Cell panel) or triplicate (Memory T Cell panel and Regulatory and γδ T Cell panel) in 96-well round-bottom plates. Cells were either overnight stimulated with ASFV-G at MOI 0.5 or incubated with media (RPMI supplemented with 2% FBS, 1× antibiotic/antimycotic, 1× non-essential amino acids (NEAA) and 1× L-glutamine) at 37 °C 5% CO_2_. The following morning, staining procedures began immediately for the Myeloid Cell panel. For the two T Cell panels, brefeldin A (1:1000) (BD, Franklin Lakes, NJ, USA) was added to the unstimulated and virus-stimulated replicates, while Leukocyte Activation Cocktail with GolgiPlug (1:333) (Southern Biotech, Birmingham, AL, USA) was added as positive control replicate and incubated for another 4 h at 37 °C 5% CO_2_. After being stained with LIVE/DEAD Fixable yellow viability dye (Invitrogen, Waltham, MA, USA), cells were staining with either the two T cell panels, memory T cell panel (Appendix A) and regulatory and γδ T Cell panel (Appendix A), following intracellular cytokine staining protocol or with the myeloid cell panel (Appendix A) following extracellular flow cytometry staining protocol. All plates were run on an Agilent NovoCyte 3000 with NovoSampler Pro System (violet, blue and red lasers) and data was analyzed in NovoExpress Software version 1.5.0.

### 4.3. Statistical Analysis

Mantel-Cox analysis was applied to survival data following challenge. Simple linear regression was applied to post-challenge log-transformed viremia and rectal temperature data to determine whether the average slope of ASFV-G-ΔI177L-vaccinated swine was significantly different than that of mock controls. Two-way analysis of variance (ANOVA) with post hoc followed by Šidák correction for multiple comparisons was applied to log_10_-transformed cytokine data. Comparisons of cell populations and median fluorescence intensity from flow cytometric analyses were analyzed by 2-way ANOVA followed by Šidák correction for multiple comparisons. For samples where zero events of the population of interest were recorded in either the unstimulated or ASFV-G-stimulated well, no MFI was calculated for that sample, resulting in some missing data. Statistical analyses were completed using GraphPad Prism version 9 (Dotmatics, Boston, MA, USA).

## Figures and Tables

**Figure 1 pathogens-11-01438-f001:**
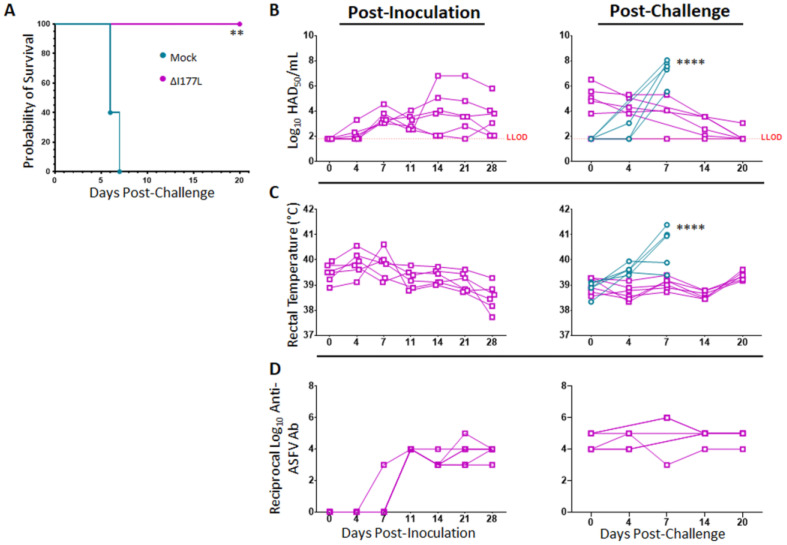
Clinical Response and Serology Following Inoculation with ASFV-G-ΔI177L and Challenge with ASF-G. At 0 dpi, swine were intramuscularly inoculated with 10^6^ median hemadsorbing dose (HAD_50_) of ASFV-G-ΔI177L or vehicle control. At 28 days post-inoculation (dpi) all animals were challenged by intramuscular injection of 10^2^ HAD_50_ of ASFV-G and monitored for up to 20 days post-challenge (dpc). (**A**) Kaplan–Meier survival curve following challenge. ** *p*-value = 0.0011 (**B**) Serum viral titers were measured by HAD_50_ at various post-inoculation and post-challenge time points, and titers are displayed in Log_10_. Difference in slope of average titer change over time was assessed by linear regression. **** *p*-value < 0.0001 (**C**) Rectal temperatures were measured at various post-inoculation and post-challenge time points and are displayed in degrees Celsius. Difference in slope of average temperature change over time was assessed by linear regression. **** *p*-value < 0.0001 (**D**) Serum anti-ASFV antibody titers were measured by ELISA at various time points during the post-inoculation and post-challenge period among ASFV-G-ΔI177L vaccinated group and are expressed as the reciprocal Log_10_ titer. Anti-ASFV antibodies have been assessed in the past among mock-vaccinated swine following inoculation and challenge and do not rise above the limit of detection prior to the humane endpoint. *n* = 5–6 swine/treatment/time point. Mock-vaccinated swine shown in blue and ASFV-G-ΔI177L-vaccinated swine shown in purple.

**Figure 2 pathogens-11-01438-f002:**
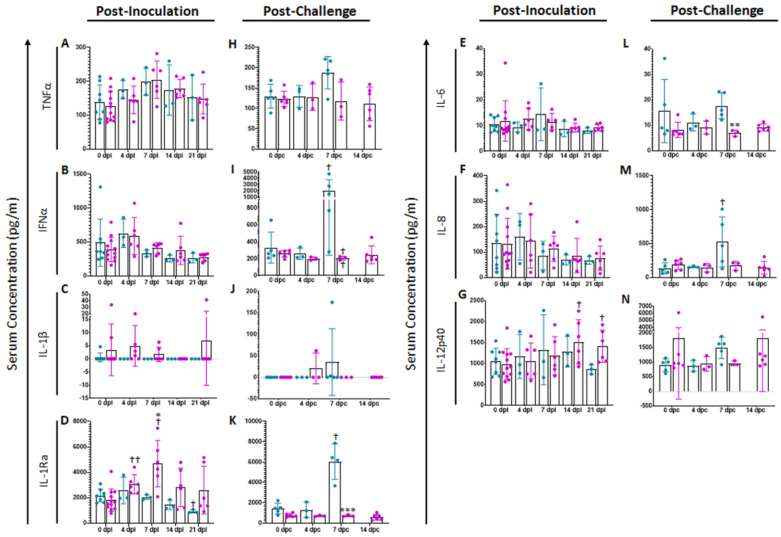
Serum Cytokine Concentrations. At various time points following inoculation (**A**–**G**) and challenge (**H**–**N**), blood was collected, allowed to clot, centrifuged and the serum collected and assayed for a variety of cytokines. *n* = 2–12 swine/treatment/time point. Mock-vaccinated swine shown in blue and ASFV-G-ΔI177L-vaccinated swine shown in purple. ** *p*-value < 0.01, *** *p*-value < 0.001 when compared with time-matched control. † *p*-value < 0.05 †† *p*-value < 0.01 when compared against treatment-matched baseline at 0 dpi.

**Figure 3 pathogens-11-01438-f003:**
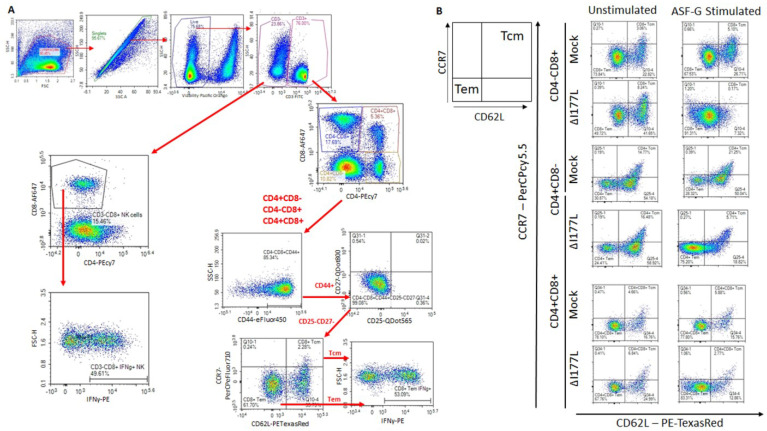
Memory T Cell and NK Cell Panel Following Inoculation. At various time points, heparinized blood was collected, PBMCs isolated and stained for flow cytometric analysis. (**A**) Gating strategy for this panel showing ultimate isolation of NK cells (CD3−CD4−CD8+), CD4+, CD8+ and CD4+CD8+ Memory T cells and IFNγ expression within those subsets. In the case of CD4+, CD8+ and CD4+CD8+ T cells, these were each subjected to CD44+CD25−CD27− identification of the memory subset and then plotted for their expression of CD62L vs. CCR7 in order to determine central memory T cells (CD62L+CCR7+) as well as effector memory T cell (CD62L−CCR7−) before analyzing the IFNγ expression for each memory population. Briefly, a generous gate was drawn around the lymphocyte population in FSC-H vs. SSC-H, then singlets were selected from a SSC-A vs. SSC-H plot, live cells were selected by gating around Viability negative cells. Expression of CD4 vs. CD8 was then assessed among CD3− as well as CD3+ cells. CD4−CD8+ were then selected from the CD3− population and assessed for IFNγ expression as NK cells. Each of three groups of T cells—CD4+CD8−, CD4−CD8+ and CD4+CD8+—were then each subjected to the same gating strategy to identify the memory subset of each as indicated above. Gates were determined by fluorescence minus one (FMO) analysis. (**B**) Comparison of CD62L and CCR7 expression on ex vivo unstimulated (left column) or ASFV-G-stimulated (right column) CD8+ and CD4+ T cells from a representative Mock- or ASFV-G-ΔI177L-vaccinated swine. *n* = 2–12 swine/treatment/time point.

**Figure 4 pathogens-11-01438-f004:**
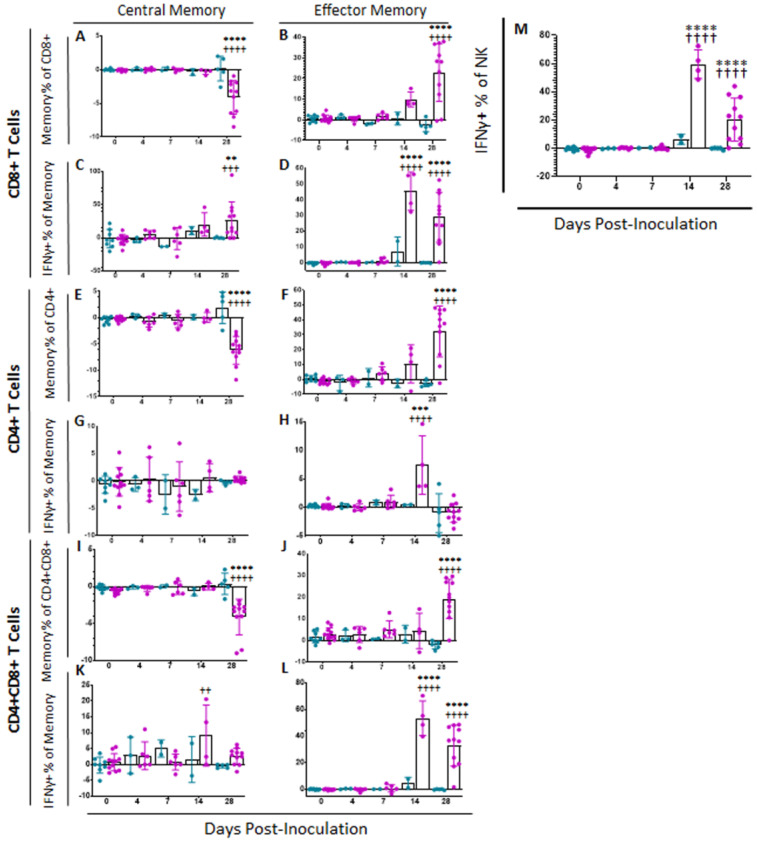
Memory T Cell and NK Cell Populations Following Inoculation. At various time points following inoculation, heparinized blood was collected, PBMCs isolated and stained for flow cytometric analysis. Upon ex vivo stimulation with ASFV-G, the change (ASFV-G-stimulated well minus unstimulated well) in CD8+ (**A**) Central (T_cm_) and (**B**) Effector memory T (T_em_) cells were calculated as a percentage of the CD4−CD8+ population, and IFNγ+ CD8+ (**C**) T_cm_ and (**D**) T_em_ cells were calculated as a percentage of the memory cell parent population. Additionally, the change (ASFV-G-stimulated well minus unstimulated well) in CD4+ (**E**) T_cm_ and (**F**) T_em_ cells were calculated as a percentage of the CD4+CD8− population, while IFNγ+ CD4+ (**G**) T_cm_ and (**H**) T_em_ cells were calculated as a percentage of the memory parent population. Finally, the change (ASFV-G-stimulated well minus unstimulated well) in CD4+CD8+ double positive (**I**) T_cm_ and (**J**) T_em_ cells were calculated as a percentage of the CD4+CD8+ population, while IFNγ+ CD4+CD8+ (**K**) T_cm_ and (**L**) T_em_ cells were calculated as a percentage of the memory parent population. (**M**) Upon ex vivo stimulation with ASFV-G, the change (ASFV-G-stimulated well minus unstimulated well) in IFNγ+ NK cells as a proportion of NK cells was also calculated. Mock-vaccinated swine shown in blue and ASFV-G-ΔI177L-vaccinated swine shown in purple. *n* = 2–12 swine/treatment/time point. ** *p*-value < 0.01 *** *p*-value < 0.001 **** *p*-value < 0.0001 when compared with time-matched control. †† *p*-value < 0.01 ††† *p*-value < 0.001 †††† *p*-value < 0.0001 when compared against treatment-matched baseline at 0 dpi.

**Figure 5 pathogens-11-01438-f005:**
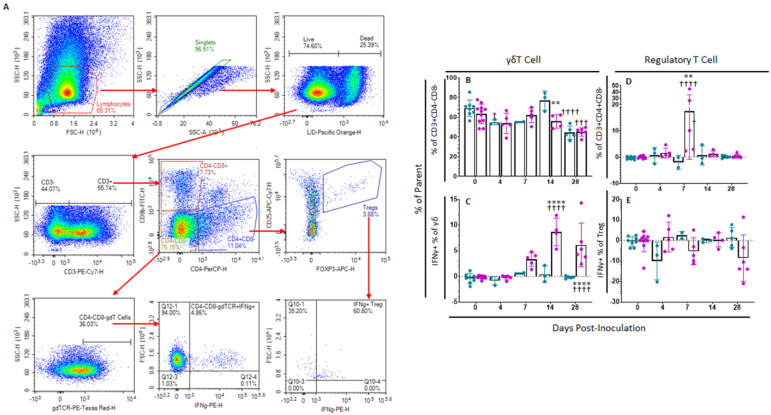
γδT cell and Regulatory T cell Panel Following Inoculation. At various time points isolated PBMCs were stained for flow cytometric analysis. (**A**) Gating strategy for this panel showing ultimate isolation of γδT cells and T_reg_, along with IFNγ expression within those subsets. Gates were determined by fluorescence minus one (FMO) analysis. (**B**) γδT cells (unstimulated well) as a percentage of their parent population were assessed among ex vivo unstimulated cells. (**C**) Change (ASFV-G-stimulated well minus unstimulated well) in IFNγ+ γδT cells as a percentage of the parent population following ex vivo stimulation with ASFV-G. Change (ASFV-G-stimulated well minus unstimulated well) in (**D**) T_reg_ cells and (**E**) IFNγ+ T_reg_ cells as a percentage of their parent population were assessed following ex vivo stimulation with ASFV-G. Mock-vaccinated swine shown in blue and ASFV-G-ΔI177L-vaccinated swine shown in purple. *n* = 2–10 swine/treatment/time point. ** *p*-value < 0.01 **** *p*-value < 0.0001 when compared with time-matched control. † *p*-value < 0.05 ††† *p*-value < 0.001 †††† *p*-value < 0.0001 when compared against treatment-matched baseline at 0 dpi.

**Figure 6 pathogens-11-01438-f006:**
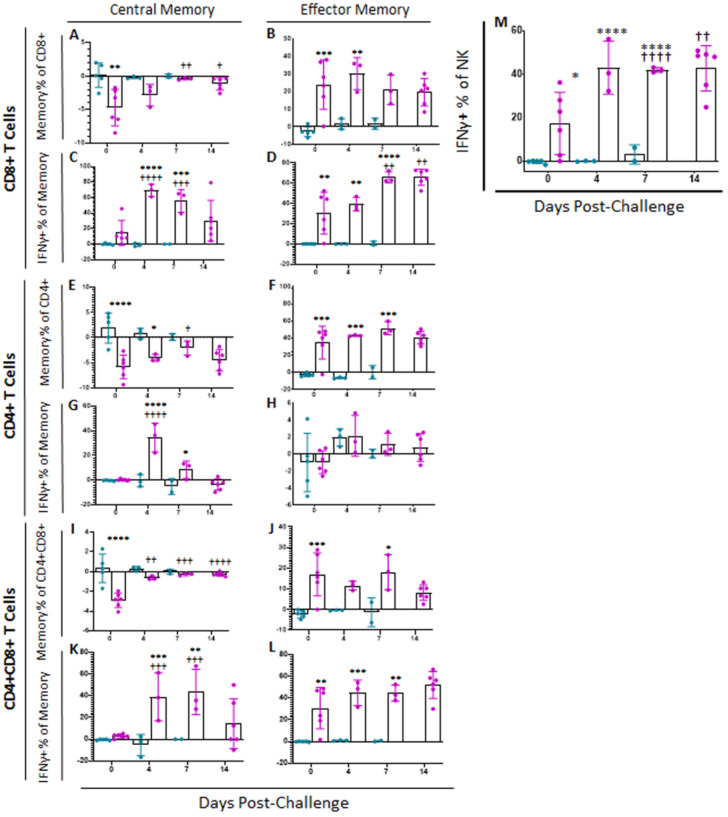
Memory T Cell and NK Cell Populations Following Challenge. At various time points following challenge, heparinized blood was collected, PBMCs isolated and stained for flow cytometric analysis. Upon ex vivo stimulation with ASF-G, the change (ASFV-G-stimulated well minus unstimulated well) in CD8+ (**A**) Central (T_cm_) and (**B**) Effector memory T (T_em_) cells, along with the IFNγ+ CD8+ (**C**) T_cm_ and (**D**) T_em_ cells were calculated as a percentage of the parent population. Additionally, the change (ASFV-G-stimulated well minus unstimulated well) in CD4+ (**E**) Central (T_cm_) and (**F**) Effector memory T (T_em_) cells, along with the IFNγ+ CD4+ (**G**) T_cm_ and (**H**) T_em_ cells were calculated as a percentage of the parent population. Finally, the change (ASFV-G-stimulated well minus unstimulated well) in CD4+CD8+ (**I**) Central (T_cm_) and (**J**) Effector memory T (T_em_) cells, along with the IFNγ+ CD4+CD8+ (**K**) T_cm_ and (**L**) T_em_ cells were calculated as a percentage of the parent population (**M**) Upon ex vivo stimulation with ASF-G, the change (ASFV-G-stimulated well minus unstimulated well) in IFNγ+ NK cells as a proportion of NK cells was also calculated. Mock-vaccinated swine shown in blue and ASFV-G-ΔI177L-vaccinated swine shown in purple. *n* = 2–12 swine/treatment/time point. * *p*-value < 0.05 ** *p*-value < 0.01 *** *p*-value < 0.001 **** *p*-value < 0.0001 when compared with time-matched control. † *p*-value < 0.05 †† *p*-value < 0.01 ††† *p*-value < 0.001 †††† *p*-value < 0.0001 when compared against treatment-matched baseline at 0 dpi.

**Figure 7 pathogens-11-01438-f007:**
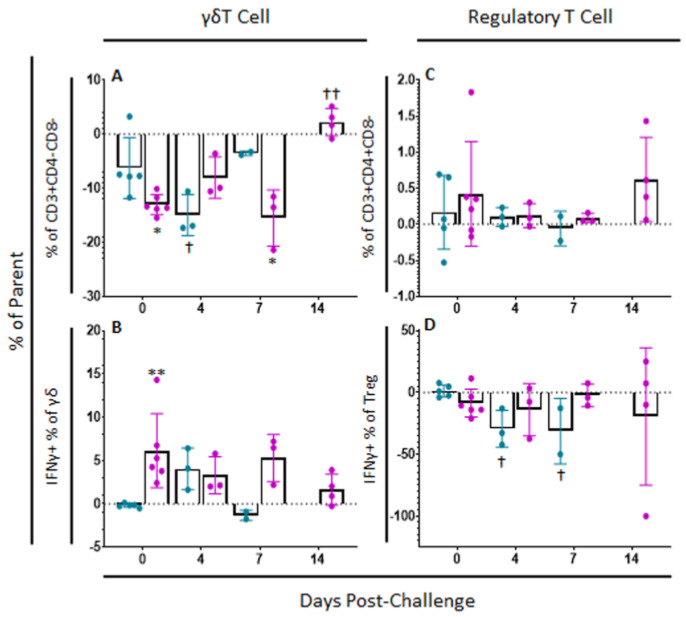
γδT cell and Regulatory T cell Panel Following Challenge. At various time points, heparinized blood was collected, PBMCs isolated and stained for flow cytometric analysis. (**A**) γδT cells (unstimulated well) as a percentage of their parent population were assessed among cells ex vivo unstimulated. (**B**) Change (ASFV-G-stimulated well minus unstimulated well) in IFNγ+ γδT cells as a percentage of their parent population following ex vivo stimulation with ASFV-G. Change (ASFV-G-stimulated well minus unstimulated well) in (**C**) T_reg_ cells and (**D**) IFNγ+ T_reg_ cells as a percentage of their parent population were also assessed following ex vivo stimulation with ASFV-G. Mock-vaccinated swine shown in blue and ASFV-G-ΔI177L-vaccinated swine shown in purple. *n* = 2–6 swine/treatment/time point. ** *p*-value < 0.01 when compared with time-matched control. † *p*-value < 0.05 †† *p*-value < 0.01 when compared against treatment-matched baseline at 0 dpi.

## Data Availability

Not applicable.

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
