# Peer review of "A Highly Effective African Swine Fever Virus Vaccine Elicits a Memory T Cell Response in Vaccinated Swine"

_pathogens, 2022, doi:10.3390/pathogens11121438_

Round 1

Reviewer 1 Report

The author's present data on a potential African Swine Fever vaccine based on a highly virulent isolate from Georgia.  Prior work has demonstrated that the G I177L is attenuated and efficacious as a vaccine.  The current work focuses on defining the immunological characteristics and timing of response following vaccination and exposure to wild-type virus. Overall, the work is interesting and important.  The experiments are well-designed, and the manuscript well written.  Swine were exposed to the vaccine candidate, periodically sampled, exposed to wild-type virus and serially sampled until 14 days post-exposure.  The ultimate goal appears to be determining the immunological response that leads to protection.   One major conclusion is that memory cells develop and this conclusion is supported by the data, though no mechanistic experiments blocking memory cell function are included and would be technically difficult to properly control.  Memory responses have not been characterized for other vaccines, so the work represents a potential next-step in ASF vaccine development. As such, mechanistic experiments should be included in the discussion.

One point of confusion is the presence of the vaccine virus at the exposure time.  What evidence do the authors have that suggest that the efficacy isn't a combination of competition between the vaccine and the wild-type viruses and an adequate memory response?

The manuscript should be proofread again, there's a few places where the meaning may be lost.

Line 271 As structured, the sentence suggests that the authors are "innate lymphoid cells..."

Line 371, first words of the paragraph are missing.

Author Response

Pathogens, Editorial Board

October 29th, 2022

Dear Dr. Penrith:

The Authors would like to thank all the reviewers for their thoughtful and helpful feedback. We believe we have addressed all the comments brought by all the reviewers in the generous time frame allotted to us.

For Reviewer 1, our edits are as follows:

  • Your point about including mechanistic experiments is well-taken and would be a thoughtful future direction for ASF research. Therefore, we have included a section covering this in the discussion at line 752 with track changes or 719 without track changes.
  • It is true that at the time of challenge, some of the swine retained a detectable amount of vaccine-strain viremia. However, as described previously, the vaccine strain viremia continues to decrease after exposure to challenge virus, and presence of wild type virus is not detectable at the vaccine dose used in this study. Therefore, we understand that the challenge virus replication is blocked at the inoculation site by a previously stimulated memory response—probably a combination of cellular and humoral, although we cannot conclude that with the results presented in the current study. We have included a sentence in the discussion to clarify this issue at line 593 with track changes or 563 without track changes.
  • We have corrected the grammatical error at original line 271—thank you for the great catch!
  • At original line 371, we did not find any words missing from the manuscript, but we have given the entire manuscript another detailed proofreading to ensure no other grammatical or formatting errors were made.

Reviewer 2 Report

Attreed et al investigate the immune response in pigs following experimental immunization with the ASFV vaccine candidate Delta-I177L and after subsequent challenge infection. A strong focus is placed on phenotypic changes and IFN-gamma production in T and NK cells following in vitro restimulation. But the authors also investigate ASFV-specific IgG antibodies and inflammatory cytokines in serum.

The study is based on a well-designed animal experiment and the data is adequately described. However, the entire flow cytometry data presented in Figures 3 to 7 is substantially impaired by several flaws.

Memory T cell and NK cell panel, Fig. 3A.
1. The CD4/CD8alpha defined phenotypes presented after separation of CD3+ and CD3negative cells are not plausible. CD3negative cells should consist of CD4negCD8alpha+ NK cells and double negative B cells but there shouldn’t be CD4+ cells (apart from a few potentially remaining pDCs). But the raw data presented indicate substantial proportions of both CD4 single positive and CD4+CD8alpha+ cells, accounting together for ~70% of all CD3neg cells. This is in contradiction to published data (e.g. doi:10.1016/j.dci.2008.06.003, Fig. 1) and also after overnight restimulation I am not aware of such phenotypes (e.g. doi:10.1128/CVI.00415-13, Fig. 3A). This puts severe doubts on the data which the authors assign to NK cells (Figs. 4I and 6I). I speculate that the gated CD4negCD8alpha+ population consists not only of NK cells but also CD8 and GammaDelta T cells. This might also explain why such a strong IFN-gamma response following ASFV restimulation was found from 14 dpi onwards. Such a classical in vitro recall response following in vivo priming with this time course is highly untypical of NK cells. Also, due to the odd CD3neg phenotypes, probably a substantial number of relevant CD4 T cells are excluded from the analysis.

2. Why do the authors exclude CD3+CD4+CD8alpha+ T cells from their analyses? A huge amount of literature shows that this phenotype consists of activated and memory CD4 T cells in swine (see for example reviews doi:10.1016/j.dci.2008.06.003 and doi: 10.1016/j.molimm.2021.04.004 which point out to original literature on this).

 3. Instead, the authors propose (and use) CD44 as a memory marker. However it was reported long ago that CD44 is expressed by most swine lymphocytes (Yang & Binns 1993,Cell Immunol.149,117e129) and that expression levels are very similar among conventional T cells, regardless of a naïve or antigen-experienced state (doi: 10.1016/j.dci.2017.07.006, Fig. 2). Considering these shortcomings, I assume that the presented CD62L+CCR7+ cells contain not only Tcm but also naïve T cells.

 4. Based on the applied phenotyping and gating, the CD8 T cells identified in Fig 3A (which built on a CD3+CD4negCD8alpha+ phenotype), most probably contain also a considerable amount of GammaDelta T cells, since these cells also express CD8alpha (see 10.1016/j.molimm.2021.04.004 chapter 2.1 for further details on this).

 5. I can’t see the relevance of applying quadrants in the bottom panel of dot plots presented in Fig. 3A when IFN-gamma is plotted against FSC-H. Apparently the percentages of these quadrants were not used for the subsequent quantitative analysis? No explanation is given.

 GammaDelta and Treg panel, Fig. 5A
1. The GammaDelta staining presented in the lower left of Fig. 5A looks inadequate. MAb clone PGBL22A should identify porcine GammaDelta T cells as a clearly separated positive population (e.g. doi:10.1016/j.dci.2008.06.003, Fig. 1E). This might be somewhat impaired by in vitro cultivation overnight, but not to such an extent. Hence, I think the GammaDelta cell data presented in Fig. 5B+C and 6A+B is inaccurate.

 2. Assuming that the data presented in Fig. 5A underwent the same in vitro stimulation as the data in Fig. 3A (potentially even from the same pig?), I find it astonishing that here CD3+CD4+CD8alpha-neg cells account for only ~11% whereas in Fig. 3A the same phenotype reaches ~54%. This puts further doubts on the presented flow cytometry data.

Finally, I can’t see how the authors identified conventional DCs from the applied marker panel for myeloid DCs. This requires additionally CADM-1 (e.g. doi: 10.4049/jimmunol.1600672).

Hence, given the strong focus of this manuscript on the T and NK cell response, I can’t recommend it for publication. Nevertheless, the applied IFN-gamma analyses are probably correct, and I apologise if the following sounds patronising. But I suggest that the authors speak to someone with experience in the analysis of porcine T cells (or study the available literature) and completely scrutinise their flow cytometry data. It might still be possible to draw meaningful conclusions from it, but this should be in a completely rewritten, newly submitted manuscript.

 Further points

Fig. 1B-D and line 118: Why is only data from 5-6 pigs shown? Prior to challenge infection both groups consisted of 10-12 pigs according to M&M, line 672. I recommend showing data from all pigs in the Delta-I177L group.

The same applies to Fig. 2: why is there such a variability in the number of animals shown per time point? For example, only data from 3 pigs is shown, e.g. mock-vaccinated pigs, 4, 7 and 14 and 21 days post inoculation and 4 days post challenge; although the figure legend in line 163 claims “n=5-12 swine/treatment/ time point”. Were not all pigs bled at all time points? If so, why?

Also in Figs. 4, 5B, 6 and 7 variable animal numbers are shown. Why? I acknowledge that placebo treated, challenged pigs were culled from 6 dpc onwards, but also before challenge there are variations in animal numbers.

Fig. 2: on both y-axis labels the ‘l’ is missing in ‘ml’.

M&M, line 673, the ‘vehicle control’ should be described in more detail.

Line 677: ‘dpc’ not ‘dpi’.

Supplementary Tables 1-3:
1. Please give in the first column of each table also the bandwidth of the respective bandpass filter used to analyse the individual fluorochromes

2. Please mention in the third column of each table also the clone name of each mAb, product numbers from companies may change over time.

3. Please mention in the last column of the three Suppl. Tables that the listed products are conjugation kits (e.g. in the header of the column).

Author Response

Pathogens, Editorial Board

October 29th, 2022

Dear Dr. Penrith:

The Authors would like to thank all the reviewers for their thoughtful and helpful feedback. We believe we have addressed all the comments brought by all the reviewers in the generous time frame allotted to us.

For Reviewer 2, our edits are as follows:

  • We agree with the reviewer that the original plots for separation of the different cell populations were not very clear. We have carefully re-run FMOs and compensation for our multichannel panels and have re-gated and re-analyzed everything obtaining the same main conclusions of the manuscript. Below we explain point-by-point why we think our data is not different from what it has been published in the past:
    • With respect to the separation of CD3+ and CD3- populations, we feel confident that our antibody and panel is separating the two populations properly as reflected in new Figure 3.
    • The new plot for NK cells shows them as CD3-CD8+, and even though there is a small spillover of CD4+ cells on the CD3- cells as compared to what was previously shown in Gerner at al., 2009, we could explain that with the fact that in the cited paper the authors were using only 4 antibodies per panel, while our panel consisted of 10 antibodies. Therefore, compensation for more antibodies may have created the spillover. Interestingly, with this new gating strategy re-applied to all the animals and timepoints, the results over the IFN secretion on NK cells (CD3-CD8+) at the different timepoints analyzed has not changed. The “adaptive” type of response observed in NK cells diverges from what is generally expected, as is pointed out by the reviewer, who cites Franzoni et al., 2013 as reference. However, Franzoni et al. are evaluating the response to another swine virus, in this case CSF, which is an RNA virus rather than a large DNA virus like ASF. Furthermore, the results presented by the mentioned group are from animals that had been vaccinated 5 days prior to challenge, and therefore results are not exactly comparable to our results. It is possible that the response of the NK cell population observed in our study would be related to what is now known as “trained immunity”. In this sense, it is now understood that while NK cells are not antigen specific, pre-exposure to a live attenuated pathogen can cause epigenetic changes that result in altered responses upon re-stimulation. To address this controversy of our results we have included a paragraph in the discussion pointing out the rarity of the results and mentioning “trained immunity” (with corresponding references) as a possible explanation that would merit further research (line 630 with track changes or 600 without track changes).
    • In the case of CD3+CD4+CD8+ cells, we agree with the reviewer that this population represents an important subset of T cells in swine, and we have now included results for these cells before and after challenge (Figures 4 and 6 and Supplementary Figures 1 and 3 and corresponding description in the Results section).
    • With respect to the concern regarding the use of CD44 for the selection of swine T memory cells, we based our panel design on two manuscripts (Franzoni et al., 2013 and Reutner at al., 2013), and therefore included CD3, CD4, CD8, CD44, CD25, CD27, CD62L and CCR7 markers to characterize the T cell populations that specifically produce IFNγ. Franzoni et al. described that CD8 T cells specifically producing IFNγ against CSFV antigen were CD44high, which lead us to use the gating strategy to analyze our data presented in the previous version of the paper. We understand the reviewers concern about the possibility of having some naïve CD8 T cells within that population. Therefore, in the new version of the manuscript we have reanalyzed the data with a new gating strategy selecting for CD27-CD25- over CD44high, to make sure that we have activated T memory cells (Reutner et al., 2013). Subsequently, this population is then analyzed for the expression of CD62L/CCR7 to select for central versus effector memory, as now depicted in the new Figure 3. We hope this new analysis strategy, more in line with what the literature has described for swine T memory cells, is sufficient to ensure that naïve cells are excluded from the population analyzed.
    • Based on the new FMOs and compensation applied, we believe that this concern about CD8 T cells containing γδ T cells as well does not apply anymore. We hope the reviewer agrees with us.
    • We agree with the reviewer’s comment regarding the quadrants and have modified the figure to better reflect the strategy used to calculate percentages of IFNγ+ cells that were applied in the subsequent graphs throughout the manuscript.
  • We understand the concern that the reviewer has with respect to the γδ T cells by analyzing the plots presented in Figure 5.
    • The antibody used to detect γδ T cells was clone PGBL22A, as previously described by others, and we have run rigorous FMOs to determine the threshold of positive and negative populations of each of our antibodies in the panel. In fact, if we were to show the plot of a naïve animal, never exposed to a vaccine but stimulated overnight with ASF antigen in Figure 5 instead, the image presented would be of a very clear positive and negative population, much more in line with what is described by Gerner et al., 2009, as you can see here: (see image in the attachment file)

              However, when the animal plotted is a vaccinated animal, whose cells are stimulated overnight with the same antigen as the vaccine, the image is different, since overnight stimulation may have an impact of the expression of the receptor. For Figure 5, if we wanted to show some positivity on IFNγ after overnight stimulation we had to choose a sample of a vaccinated animal and prioritize that versus a naïve animal with a plot with a clearer separation of the γδ T cell population. In any case, the gates for the analysis were stablished at the ~103 fluorescence intensity mark for the appropriate channel, and were applied to all the samples (animals and time points) the same way, so the data should be reliable.  

  • The panel of antibodies used to create the plots in Figures 3 and 5 are not the same (please compare Supplementary Tables 1 and 2) and that may have an impact on the exact percentages of the different populations. Additionally, we did not use the same animal for these two representative gating strategies. However, with the new FMOs and compensations applied to the panel shown in Figure 3, the percentages of single positive CD4 and CD8 cells are now more aligned. We would also like to clarify that the staining done to analyze T cell memory was applied to fresh cells, while the staining to analyze the γδ T cell and Treg panel was applied to revived cryopreserved cells, so the overall number could be expected to be slightly different. With regards to the CD3 positive and negative cell population separation for Figure 5, we would like to clarify that, for the figure, we have again chosen to show overnight stimulated cells from a vaccinated animal to be able to detect IFN γ+ cells. When unstimulated cells from a control animal at this time point are plotted for CD3, the image shown reflects the separation of the populations much better: (see image in attachment file)

As explained for Figure 3, the gating threshold has been set up at 103 for all plots based on our FMO, so the analysis should be accurate for all time points/animals/stimulation status.

  • We have based our panel for the myeloid cells (monocytes and dendritic cells) on a recent publication, Sanchez-Cordon et al., 2020. We understand that, based on Auray et a., 2016, the reviewer would like to have CADM1 marker included in the panel to analyze the dendritic cell populations of the manuscript. However, since the results obtained on the analysis of these populations did not show anything of great importance and do not have a great impact on the overall conclusions of the manuscript, we have decided to maintain the panel as is and describe the populations based on the markers expression of the populations instead of defining them as conventional or plasmacytoid dendritic cells. We have formatted the text of the manuscript and the tables to reflect that accordingly.

We hope that all these changes and the re-analysis performed have fulfilled expectations of the reviewer and it is now an acceptable version to be published in Pathogens.

Further points brought up by Reviewer 2

  • With respect to the number of animals used in this study, 5 placebo and 6 ASFV-G-ΔI177L swine were assessed during the pre-challenge timepoints, including 28 dpi. These specific individual swine were then retained for challenge at a later timepoint that was not covered in this paper. The swine that were challenged at 28 dpi were a separate set of 5 placebo and 6 ASFV-G-ΔI177L-inoculated swine that were inoculated and sampled at the same 0 dpi timepoint as the other set, and then not sampled again until 28 dpi when they were challenged. This is related to the overall more complex study design aiming to answer more questions than the ones presented in this manuscript. That larger study will be the subject of a future manuscript by our research group. Due to logistical difficulties related to working in a BSL3-Ag facility and in accordance with our IACUC guidelines, which suggest reducing the number of procedures performed in swine to the minimum possible, only some of the groups were schedule to be sampled. Also, in accordance with IACUC guidelines, each individual swine may not have blood drawn more frequently than every 3 days and a pig must be removed from bleeding schedules if they develop hematomas during bleeding procedures. Therefore, a subset of animals in each group was sampled at 4 and 7 dpc, explaining the different numbers observed in various figures. For MFI plots specifically, if either the unstimulated or ASFV-G-stimulated well produced zero cells of a particular population, the change in MFI was not calculated, resulting in further reduced numbers in some of these plots. The authors agree that these inconsistencies could have been explained more clearly and have added language at lines 788 and 862 with track changes or 755 and 829 without track changes in the Materials and Methods.
  • Figure 2 has been formatted accordingly.
  • The vehicle control has been described in the Material and Method section (line 783 with track changes or 750 with track changes).
  • Line 677 has been corrected as suggested.
  • All requests related to Supplementary Tables 1-3 have been addressed as suggested.

New version of manuscript in attachment

Reviewer 3 Report

ASFV continuous spread and thread the global pig industry and food security. Vaccine is the preferred method to control ASF. Understanding the immune response mechanism for protection is critical for vaccine development and evaluating efficacy of developed vaccines. This manuscript assessed and compared the humoral and cellular immune responses of ASFV-G-ΔI177L vaccinated and control pigs. Overall, this manuscript is well-written and well-organized. I recommend to publish this manuscript in pathogens Journal.

      Comment: This manuscript can be improved by adding the neutralizing antibody data.

Author Response

Pathogens, Editorial Board

October 29th, 2022

Dear Dr. Penrith:

The Authors would like to thank all the reviewers for their thoughtful and helpful feedback. We believe we have addressed all the comments brought by all the reviewers in the generous time frame allotted to us.

For Reviewer 3, our edits are as follows:

  • This is a well-taken point about the importance of neutralizing titers versus ELISA-assessed titers. However, we regret that an assay to measure this endpoint does not currently exist, but we are working hard on our end to try to develop it. As an example of that, some authors on this manuscript recently published an alternative method to demonstrate neutralization based on the use of flow cytometry (Canter et al. 2022). However, the technique they developed used a different variant of virus, so it would need to be adapted to the virus of interest for the current study.

New version of manuscript in attachment.

Reviewer 4 Report

The manuscript by Attreed and colleagues is very timely since it characterizes the immune responses of pigs following vaccination with a recently approved live attenuated vaccine against African Swine Fever (ASF).  A very thorough evaluation of different immune cell populations is presented, and this may contribute to our understanding of mechanisms of protection, which remain elusive for ASF. Unfortunately, the vaccine dose used in the current study (106 HAD50) is much higher than the one used in vaccine trials (102.6 HAD50), for example in here:  https://doi.org/10.3390/v14050896. This precludes any major conclusions regarding immune responses since these may be dose dependent. Another major drawback of the manuscript is the failure to explain the rationale for presenting the data on cell populations as differences in percentages between cells stimulated with ASFV and unstimulated cells. Changes in different cell populations purified from blood from vaccinated and placebo groups should be visible in non-stimulated cells. It is not clear why some cell populations increase or decrease following stimulation. Cell proliferation would probably take longer than 24 hours, the time used to stimulate the cells. On the other hand, cell death would cause changes in percentages of bystander cells too. Is ASFV “stimulation” causing downregulation of the markers CD62L and CCR7? This needs to be discussed. I would suggest the authors to show both data sets (unstimulated and stimulated cells) and not the “normalized” results (stimulated/unstimulated). This of course is not needed for the IFNγ results since one expects that only after stimulation a significant increase would be noticeable.

Other points:

Line 17: Please remove “identification of” from the sentence

Line 129-130: The cytokine storm is not well characterized. As the authors show in figure 2, levels of cytokines are not that high after challenge even in non-vaccinated pigs.

Figure 3: Please use the same annotation in the panels and figure legends (CCR7 instead of CD197)

Lines 638-641: These observations are not correlates of protection, since as explained above the vaccine dose may have impacted on the results. Additionally, a correlate of protection would be better defined in a group of pigs immunized with the same vaccine/dose but with different outcomes.

Author Response

Pathogens, Editorial Board

October 29th, 2022

Dear Dr. Penrith:

The Authors would like to thank all the reviewers for their thoughtful and helpful feedback. We believe we have addressed all the comments brought by all the reviewers in the generous time frame allotted to us.

For Reviewer 4, our edits are as follows:

  • Regarding the dose of the vaccine used in the present study, this is a replication of a dose that was administered in the following paper by our group (https://doi .org/10.1128/JVI.02017-19). While the Tran paper (doi.org/10.3390/v14050896) cited by the Reviewer was a more recent safety and efficacy study of this particular deletion mutant, the 2020 paper by our group establishes 102 HAD50, 104 HAD50 and 106 HAD50 as all being within the safe therapeutic range of this particular attenuated virus. Still, the reviewer’s point about the dose dependency of immunological responses is well-taken and has been addressed in Line 746 with track changes or 713 without track changes.
  • Regarding population shifts following ex vivo restimulation, we do indeed hypothesize that CCR7 and CD62L surface expression is being downregulated subsequent to restimulation rather than proliferation. Please see Figure 3B for our visual representation of this shift. There are not differences in central and effector memory populations in the unstimulated wells between treatment groups. We have included a sentence in the results to this effect at Line 264 with track changes or line 256 without track changes. This shift occurs only upon restimulation. The reviewer’s points about changes in γδ T cells is sensible and we have therefore reanalyzed looking specifically at this population as a percent of the parent population from the unstimulated wells. We have edited Figures 5 and 7 and the associated text in the Results section. However, with respect to Treg cells, we were interested in analyzing changes of the primed Treg population after secondary exposure to the vaccine antigen. In that sense, we expect that FoxP3, the transcription factor that regulates development and function of Tregs, may change after stimulation as compared with unstimulated cells. Therefore, we are going to maintain graphs related to Treg populations analysis over stimulated cells.
  • Line 17 has been corrected. We thank the Reviewer for another good catch!
  • Line 129-130 the Authors agree that the systemic cytokine storm phenomenon has only recently been described for ASF and have modified the text accordingly (lines 130-133 of new version with or without track changes). The paragraph’s intention was to show the reader our interest in studying the cytokine profile induced by the vaccine strain and analyze possible differences between the naïve animals after challenge. As the reviewer pointed out, we do observe a less exuberant induction of cytokines in naïve animals after challenge than originally anticipated, but we still described the significant upregulation of IFNα, IL-1RA, and IL-8 and observed a non-significant trend in three other analytes, TNFα, IL-6 and IL-12p40 as compared to 0 dpc.
  • Figure 3 has been edited to harmonize the nomenclature of CCR7 throughout.
  • We understand the reviewer’s concern with regards to the language use around correlates of protection. However, as explained before, the dose used in this work had been previously used as part of the initial report of this attenuated vaccine candidate in which several doses (102 to 106 HAD50) were tested (Borca et al., 2020). Interestingly, in that study, all animals were not only protected regardless of the vaccine dose used, but also developed the same levels of specific antibodies. In the current study, we observed protection, equivalent levels of antibodies as observed before, and now described new parameters of cellular response that correlate with protection. Based on the previous results on the antibody levels, we believe that when lower doses of vaccines are used, the cellular response parameters are going to remain similar to the ones described here. However, as pointed out by the reviewer, it deserves further confirmation, and, as such, we have included a line in the discussion making that clarification Line 746 with track changes or 713 without track changes.

We would again like to extend our sincerest thanks to all 4 reviewers for their careful and thoughtful review of our manuscript. We hope that with the new version addressing all the points brought up during the review process, our manuscript will be acceptable for publication.

Thank you for your consideration, and most sincerely,

Fayna Diaz San Segundo

For new version of manuscript see attachment.

Round 2

Reviewer 2 Report

The authors resolved most of my queries, but I still have several issues that I would like to see addressed before I can recommend the manuscript for publication.

The basic phenotyping of NK cells and T cell subsets presented in Fig. 3A has improved substantially.

This brings me to two additional issues: now that my concerns on the basic T and NK cell phenotyping have been resolved I can focus with more confidence on the presented CD44, CD62L and CCR7 phenotypes. I agree with the authors that the data presented in Franzoni et al. 2013 suggest that porcine memory T cells have a CD44high phenotype. However, there are not many additional studies investigating CD44 as a memory marker in porcine T cells in more detail. For example, the work by Yang et al. 2017 (https://doi.org/10.1016/j.dci.2017.07.006) shows that ex vivo nearly all T cells have a uniform CD44high phenotype. I appreciate that the authors now aimed to investigate also CD25 and CD27 expression in this phenotyping panel, but the example presented in Fig. 3A raises several questions. According to Reutner et al. 2013, porcine CD4 Tcm should be CD27+, which does not apply to the presented example. I could not deduce from the figure legend of Fig. 3 if the data presented in (A) is from an ASF-G restimulated Delta-I177L immunised pig but assume that this is the case, considering the high percentages of IFN-gamma producing cells presented. Given that such cells seem to consist mainly of CD62LnegCCR7neg cells, this might explain why there are no CD27+ cells but some doubts remain. In addition, I would have assumed that some of these re-stimulated T cells express CD25. To clearly demonstrate the quality of the presented flow cytometry data I recommend to the authors to create for Fig. 3A a corresponding supplementary figure that shows the applied gating also for

A) unstimulated cells from a Delta-I177L pig

B) unstimulated cells from a mock pig

C) ASF-G stimulated cells a mock pig

Ideally, each of these should also show CD44, CD25, CD27, CD62L and CCR7 for all three CD4/CD8-defined T cell subsets. I also recommend showing CD44 in combination with IFN-gamma for the different stimulation and animal groups which may or may not corroborate the claim that CD44high corresponds to a memory phenotype.

 I also need to come back on the presented CD62L/CCR7 phenotypes. Based on established phenotypes from human T cells, the authors designate CD62L+CCR7+ cells as Tcm and CD62LnegCCR7neg cells as Tem. I agree to this, but the presented data show that at least in unstimulated cells and cells from mock pigs there is a substantial proportion of CD62L+CCR7neg cells, which the authors exclude form further analyses. Why? I would recommend analysing this phenotype also quantitatively in Fig. 4, including IFN-gamma production.

 Fig. 4 and legend of Fig. 4: I understand that percentages shown are calculated as the difference between the ASFV-G stimulated cultures and the corresponding unstimulated cultures (line 191-193). However, I think this this information not clearly provided in the figure legend. Please add or clarify. The same applies to the figure legends of Figure 6.

 I have the same question related to Fig. 5: now that the authors have resolved my concern about the quality of the CD3 and GammaDelta staining, I paid closer attention to the %ages shown in Fig. 5B-D. If I understand the legend correctly, for GammaDelta T cells shown in Fig. 5B, the authors show %ages of GammaDelta cells from unstimulated in vitro cultures. However, for Tregs, (Fig. 5D) it appears that the difference between the ASFV-G stimulated cultures and the corresponding unstimulated cultures has been calculated. Why did the authors choose these two different approaches? Please explain, also for Fig. 7A and 7D.

Author Response

November 16th, 2022

Dear Dr. Penrith and Reviewers:

The Authors would like to thank Reviewers 2 and 4 for their continued helpful feedback on our manuscript. We hope we have addressed all the comments brought by all the reviewers during this second round of comments.

For Reviewer 2, our edits are as follows:

  • Regarding the animal chosen for the gating strategy presented in Figure 3A, the Reviewer is correct—this was a vaccinated, ex vivo ASFV-G-stimulated individual at 28dpi. As mentioned before, the main objective of the figure is to show the plotting and gating strategy that brought us to IFNγ production upon stimulation. For that purpose, we chose an animal that would indeed respond.
  • With regards to CD44 and IFNγ expression, we do indeed see that, for the most part, only CD44+ T cells express this cytokine. Results are illustrated in the panels below [on the left is unstimulated (U) and on the right is stimulated (ASFV-G)]. [PLEASE SEE PLOTS FROM THE ATTACHMENT].           

We have included a new explanatory sentence to this effect [line 199], thereby adding to the body of knowledge about porcine immunology. However, we respectfully disagree with Reviewer 2 on the creation of a supplementary figure showing the expression patterns of CD44, CD25, CD27, CD62L and CCR7 for ASFV-stimulated and unstimulated cells for both mock and vaccinated swine for all three T cell subsets characterized in this paper, as we think that there is little to be achieved by this exercise. We have demonstrated to the Reviewer and confirmed in our manuscript that IFNγ expression mostly only attends CD44+ T cells. We acknowledge the variability in the expression of CD27 and CD25 among animals, regardless of immunization status and unfortunately, we do not see a strong correlation between CD27 positivity and IFNγ expression when compared to the analysis of Reutner et al. However, as explained before, after back gating T cell subsets, the vast majority of IFNγ positive cells are CD44High, as previously described for T memory cells by Franzoni et al., 2013. Furthermore, we differentiated between effector and central based on the expression of CD26L and CCR7, by using the same strategy described by the same authors for swine and by others for other species.

  • With regards to CD62L+CCR7- cells, we respectfully disagree with the Reviewer about analyzing these cells, as, to the best of the authors’ knowledge, these cells have not been characterized in either the human or porcine literature. While preliminary analyses show that these cells express some IFNγ upon stimulation in vaccinated animals—at a level intermediary to that of Tem and Tcm from the corresponding T cell subset—while there is no IFNγ production observed in the mock and unstimulated samples. Unfortunately, given the unknown definition of this population, we feel that adding this analysis would be outside the scope of this paper.

  • We have included a statement in each of the Figures 4 – 7 and Supplemental Figures 1 – 4 to clarify that measured parameters reflect the difference between the ASFV stimulated versus unstimulated cells. Additionally, this has been defined in the body of the manuscript text at Line 192.

  • Regarding the different approach to handling the γδT cells as compared to Treg cells, please see our prior response to reviewer 4, who called on us to take a second look at this data. In brief, we explained that the reviewer’s points about changes in γδ T cells were sensible and we therefore reanalyzed the data by looking specifically at this population as a percent of the parent population in unstimulated cells. We edited Figures 5 and 7 and the associated text in the Results section to reflect the new analyses during the first round of review. However, with respect to Treg cells, we would like to point out, as also explained to Reviewer 4 before, that we were interested in analyzing changes of the primed Treg population after secondary exposure to the vaccine antigen. With this prospective, we expected that FoxP3, the transcription factor that regulates development and function of Tregs, might change after stimulation as compared to unstimulated cells. Therefore, we consider appropriate maintaining graphs related to Treg populations analysis over stimulated cells.

    We would again like to again extend our sincerest thanks to all four reviewers for their careful and thoughtful review of our manuscript. We hope that with the new version addressing all the points brought up during the review process, our manuscript will be acceptable for publication.

    Thank you for your consideration, and most sincerely,

    Fayna Diaz San Segundo

Reviewer 4 Report

The authors responded to my concerns and the new figures substantially improved the manuscript.  

Figure 3, left lower panel: X-axis should be IFNg instead of CD4. 

Author Response

November 16th, 2022

Dear Dr. Penrith and Reviewers:

The Authors would like to thank Reviewers 2 and 4 for their continued helpful feedback on our manuscript. We hope we have addressed all the comments brought by all the reviewers during this second round of comments.

For Reviewer 4, our edit is as follows:

  • We have made the recommended change to Figure 3. Thank you again for a good catch!

We would again like to again extend our sincerest thanks to all four reviewers for their careful and thoughtful review of our manuscript. We hope that with the new version addressing all the points brought up during the review process, our manuscript will be acceptable for publication.

Thank you for your consideration, and most sincerely,

Fayna Diaz San Segundo

Round 3

Reviewer 2 Report

The authors are not willing to address my concerns regarding the analysis of their flow cytometry data. Hence, I cannot make a positive recommendation for publication.